# Unbiased Alignment for Large Language Models with Noisy Preferences

Jialiang Wang [1 2]   Xianming Liu [1]   Xiong Zhou [1]   Hui Liu [2]   Haoliang Li [2]

## Abstract

The alignment of large language models with human preferences is commonly achieved through Reinforcement Learning from Human Feedback or Direct Preference Optimization. However, these methods are vulnerable to the significant noise prevalent in real-world preference datasets. To address this critical issue, we present a theoretical framework for unbiased alignment, introducing the *Unbiased Reward Model* (URM) loss and the *Unbiased Direct Preference Optimization* (UDPO) loss. By mathematically correcting the distortion induced by preference noise, our novel objectives enable unbiased model training directly from noisy datasets, without requiring clean ground-truth supervision. We provide rigorous theoretical analyses demonstrating that our methods are noise-tolerant, parameter downward compatible, and classification-calibrated. Comprehensive experiments across diverse datasets demonstrate that our approaches outperform state-of-the-art baselines. Code available at: https://github.com/cswjl/unbiased-alignment.

## 1. Introduction

Large language models (LLMs) have achieved remarkable progress in various fields (Naveed et al., 2025; Li et al., 2025). The preference alignment following pre-training and supervised fine-tuning has become a standard stage in LLM training pipelines (Ouyang et al., 2022; Liu et al., 2024). This stage plays a crucial role in improving the model's ability to follow human preferences and enhancing its overall performance (Rafailov et al., 2023) However, real-world preference datasets often contain a significant proportion of noisy preference pairs, usually ranging from 20% to 40% (Gao et al., 2024). As illustrated in Figure 1, noisy preferences from annotators can introduce ambiguity into the

[1]Harbin Institute of Technology [2]City University of Hong Kong. Correspondence to: Xianming Liu <csxm@hit.edu.cn>.

*Proceedings of the 43rd International Conference on Machine Learning*, Seoul, South Korea. PMLR 306, 2026. Copyright 2026 by the author(s).

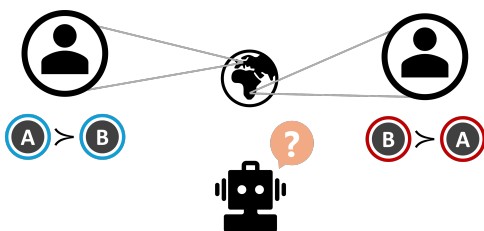

*Figure 1.* An illustration of noisy preferences in LLM alignment.

learning signal. These noisy preferences often stem from a lack of professional knowledge, human carelessness, or social bias, significantly reducing the performance and safety of LLMs (Zhang et al., 2017; Gao et al., 2024). Therefore, aligning with noisy preferences presents a critical challenge during the post-training phase of LLMs (Gao et al., 2024; Chowdhury et al., 2024; Wu et al., 2025).

While several methods for alignment with noisy preferences have been proposed recently, they often suffer from significant limitations. Specifically, these approaches either fail to achieve full noise-tolerance (Wu et al., 2025; Liang et al., 2025) or require a precise estimation of the noise rate in the dataset (Mitchell, 2023; Chowdhury et al., 2024), which is unfeasible in reality. In this work, we independently propose a unified theoretical framework for unbiased alignment. Our core insight is that preference noise follows a statistically modeled transition; by mathematically inverting this process, we can recover the unbiased model even when training solely on noisy data. We consider the two dominant paradigms for LLM alignment: Reinforcement Learning from Human Feedback (RLHF) (Christiano et al., 2017; Ouyang et al., 2022) and Direct Preference Optimization (DPO) (Rafailov et al., 2023). We separately explore how to achieve unbiased alignment of these two strategies using similar theoretical ideas.

For the RLHF paradigm, preference noise primarily affects the reward model training stage, where a noisy reward model can propagate erroneous signals to the subsequent RL phase. We characterize the distortion of reward signals caused by noisy preferences and establish a mathematical transformation relationship between unbiased and noisy reward models. Based on this analysis, we derive the *Unbiased Reward Model* (URM) loss. This loss function enables the training of an unbiased reward model directly from noisy data, effectively mitigating noise-induced bias at the first

stage of RLHF.

For the DPO paradigm, which simplifies RLHF by directly optimizing the policy without an explicit reward model, noisy preferences can also substantially bias the learned policy. We demonstrate how noise distorts the implicit reward signal and establish the transformation relationship between the unbiased and noisy policy models. Subsequently, we derive the *Unbiased Direct Preference Optimization* (UDPO) loss. This loss allows for the direct optimization of an unbiased policy model, maintaining the simplicity of DPO while imparting robust capability.

The key to the success of DPO lies in satisfying the mathematical equivalence with RLHF. We demonstrate that this equivalence extends to our proposed URM and UDPO objectives, providing a unified unbiased alignment framework. Under this framework, we conduct a rigorous theoretical analysis encompassing gradient analysis, parameter downward compatibility, classification calibration, and the excess risk bound. These results substantiate our method's robustness, parameter generalization, and its capacity to recover a Bayes optimal classifier.

Our contributions are summarized as follows:

- For the RLHF paradigm, we develop a noisy reward model correction theory and derive a robust *Unbiased Reward Model* (URM) loss.

- For the DPO paradigm, we develop a noisy policy model correction theory and derive a robust *Unbiased Direct Preference Optimization* (UDPO) loss.

- We thoroughly explore the theoretical properties of the proposed unbiased alignment framework, demonstrating the robustness and practical utility of our method.

- We conduct extensive experiments across multiple datasets and model families, empirically demonstrating the superiority of our proposed framework.

## 2. Related Work

Modern LLM training typically follows a three-stage pipeline (Ouyang et al., 2022; Bai et al., 2023; Grattafiori et al., 2024; Yang et al., 2025): 1) Pre-Training (Brown et al., 2020): The model is trained on a large-scale corpus by maximizing the likelihood of the next token. 2) Supervised Fine-Tuning (SFT) (Raffel et al., 2020; Wang et al., 2023): The pre-trained model is fine-tuned on high-quality instruction data, yielding the SFT model $\pi_{\text{sft}}$. 3) Preference Alignment (Christiano et al., 2017; Ouyang et al., 2022; Rafailov et al., 2023): The SFT model is further trained on preference data to align with human values and preferences. This work focuses specifically on the preference alignment stage.

**Preference Alignment.** RLHF (Christiano et al., 2017; Ouyang et al., 2022) is the foundational paradigm for preference alignment. It traditionally involves training a reward model from preference data, followed by policy optimization using an RL algorithm, most notably PPO (Schulman et al., 2017) and GRPO (Shao et al., 2024). However, the RLHF pipeline is often criticized for its substantial computational overhead and optimization instability. To address these challenges, DPO (Rafailov et al., 2023) has emerged as a compelling alternative that streamlines RLHF by directly optimizing the policy from preference pairs. This has inspired a growing body of work on supervised preference optimization, such as SLiC-HF (Zhao et al., 2023), IPO (Azar et al., 2024), KTO (Ethayarajh et al., 2024), SimPO (Meng et al., 2024), and DiscoPOP (Lu et al., 2024). More recently, increasing attention has been devoted to alignment under noisy preferences, which is the central focus of this paper. rDPO (Chowdhury et al., 2024) proposes a provably robust alignment objective, but it requires a precise estimation of the dataset noise rate, which is difficult to obtain in practice. Other methods, such as ROPO (Liang et al., 2025) and Dr.DPO (Wu et al., 2025), introduce alternative loss functions to mitigate the impact of noisy preferences, but they remain limited in achieving full noise-tolerance.

## 3. Preliminary

In this section, we elaborate on two mainstream paradigms for preference alignment: RLHF (Christiano et al., 2017; Ouyang et al., 2022) and DPO (Rafailov et al., 2023), which form the foundations of our work. We also introduce the noisy preference model used in our analysis.

**RLHF.** RLHF typically consists of two stages. In the first stage, a reward model $r_\phi$ is trained on a static preference dataset $\mathcal{D} = \{x^{(i)}, y_w^{(i)}, y_l^{(i)}\}_{i=1}^N$, where $y_w$ denotes the response preferred over $y_l$ for the prompt $x$, written as $y_w \succ y_l$. Under the widely used Bradley-Terry (BT) model (Bradley & Terry, 1952), human preference is modeled as:

$$p(y_w \succ y_l | x) = \sigma\big(r(x, y_w) - r(x, y_l)\big), \quad (1)$$

where $\sigma$ is the sigmoid function. Following standard practice, we assume that the ground-truth preference $p^*(y_w \succ y_l) = 1$. The reward model $r_\phi$ is optimized by maximum likelihood estimation, minimizing the following reward model (RM) loss:

$$\mathcal{L}_{\text{RM}} = -\log \sigma\big(r_\phi(x, y_w) - r_\phi(x, y_l)\big). \quad (2)$$

In the second stage, the policy $\pi_\theta$ is optimized to maximize the learned reward. To prevent the policy from deviating excessively from the reference model $\pi_{\text{ref}}$, a KL-divergence

penalty is incorporated (Jaques et al., 2017; 2020):

$$\max_{\pi_\theta} \mathbb{E}_{x \sim \mathcal{D}, y \sim \pi_\theta(y|x)} \left[ r_\phi(x, y) \right] - \beta \mathbb{D}_{\text{KL}} \left[ \pi_\theta(y|x) \| \pi_{\text{ref}}(y|x) \right],$$
(3)

where $\beta$ is a hyperparameter. In practice, both $\pi_\theta$ and $\pi_{\text{ref}}$ are initialized from the SFT model $\pi_{\text{sft}}$. Due to the discrete nature of the language generation, this objective is non-differentiable and is typically optimized using an RL algorithm.

**DPO.** DPO simplifies the RLHF pipeline by directly optimizing the policy model $\pi_\theta$ on preference data through a supervised preference objective, eliminating the need to train an explicit reward model and employ an RL algorithm. Rafailov et al. (2023) show that the optimal solution to the RLHF objective in Equation 3 can be expressed in closed form as:

$$r(x, y) = \beta \log \frac{\pi^*(y|x)}{\pi_{\text{ref}}(y|x)} + \beta \log Z(x),$$
(4)

where $Z(x) = \sum_y \pi_{\text{ref}}(y|x) \exp\left(\frac{1}{\beta} r(x, y)\right)$ is the partition function. By substituting Equation 4 into the BT model in Equation 1, the partition function $Z(x)$ cancels out. This allows for optimizing the policy via a maximum likelihood loss:

$$\begin{aligned}
\mathcal{L}_{\text{DPO}} &= -\log \sigma \left( \beta \log \frac{\pi_\theta(y_w|x)}{\pi_{\text{ref}}(y_w|x)} - \beta \log \frac{\pi_\theta(y_l|x)}{\pi_{\text{ref}}(y_l|x)} \right) \\
&= -\log \sigma \left( \beta \log \frac{\pi_\theta(y_w|x)\pi_{\text{ref}}(y_l|x)}{\pi_\theta(y_l|x)\pi_{\text{ref}}(y_w|x)} \right).
\end{aligned}$$
(5)

**Noisy Preference Model.** We assume that the samples in the dataset $\mathcal{D}^\eta$ are not drawn from the unbiased human preference distribution $\mathcal{P}^*$, but rather from a noisy distribution $\mathcal{P}^\eta$ influenced by factors such as cognitive errors and social biases:

$$p^\eta(y_w \succ y_l|x) = (1-\eta)p^*(y_w \succ y_l|x) + \eta p^*(y_l \succ y_w|x),$$
(6)

where $\eta \in [0, \frac{1}{2})$ denotes the noise rate, while $p^\eta$ and $p^*$ are the preference probabilities under the distributions $\mathcal{P}^\eta$ and $\mathcal{P}^*$, respectively.

## 4. Unbiased Alignment

Since real-world preference datasets often contain noise, standard approaches such as RLHF and DPO may struggle to achieve ideal performance. To address this challenge, we investigate the theoretical foundations of noise model correction and propose principled methods for unbiased alignment. Detailed proofs are provided in the Appendix A.

### 4.1. Unbiased Reward Model Learning

We first explore the reward model learning stage in RLHF. When preference data is contaminated with noise, the reward model exhibits a systematic bias, yielding a reward signal that is distorted by the underlying noise distribution. This relationship is characterized as follows:

**Lemma 4.1.** *Under noisy preferences, the normal reward model loss* $-\mathbb{E}_{(x,y_w,y_l)\sim\mathcal{P}^\eta}\left[\log\sigma(r_\phi(x,y_w) - r_\phi(x,y_l))\right]$ *is equivalent to the following noisy optimization loss:*

$$\begin{aligned}
- \mathbb{E}_{(x,y_w,y_l)\sim\mathcal{P}^*}\big[ &(1-\eta)\log\sigma(r_\phi(x,y_w) - r_\phi(x,y_l)) \\
&+ \eta\log\sigma(r_\phi(x,y_l) - r_\phi(x,y_w))\big].
\end{aligned}$$
(7)

*The corresponding optimal noisy reward model satisfies*

$$\begin{aligned}
\sigma(r_\phi^\eta(x,y_w) - r_\phi^\eta(x,y_l)) = \\
(1-\eta)\,p^*(y_w \succ y_l|x) + \eta\,p^*(y_l \succ y_w|x).
\end{aligned}$$
(8)

**Noisy Reward Model Correction.** To address the issue of noise, following Lemma 4.1, we demonstrate that unbiased reward signals can be derived from their noisy counterparts.

**Theorem 4.2.** *For any* $(x, y_w, y_l) \sim \mathcal{P}^*$, *if* $r_\phi^\eta(x, y)$ *is the optimal noisy reward model, then the optimal unbiased reward model* $r_\phi^*(x, y)$ *satisfies:*

$$\begin{aligned}
r_\phi^*(x, y_w) - r_\phi^*(x, y_l) = \\
\log\left[ \frac{\exp(r_\phi^\eta(x,y_w) - r_\phi^\eta(x,y_l)) - a}{1 - a \cdot \exp(r_\phi^\eta(x,y_w) - r_\phi^\eta(x,y_l))} \right],
\end{aligned}$$
(9)

*where* $a = \frac{\eta}{1-\eta}$ *is a constant.*

It is evident that the unbiased reward signal $r_\phi^*(x, y)$ can be recovered from its noisy counterpart $r_\phi^\eta(x, y)$. For instance, in the case where the noisy reward model is unbiased ($a = 0$), the relationship reduces to $r_\phi^*(x, y_w) - r_\phi^*(x, y_l) = r_\phi^\eta(x, y_w) - r_\phi^\eta(x, y_l)$, signifying an absence of confounding. Conversely, when the noisy model is completely inverted relative to true preferences (as $a \to \infty$), we obtain $r_\phi^*(x, y_w) - r_\phi^*(x, y_l) = r_\phi^\eta(x, y_l) - r_\phi^\eta(x, y_w)$, representing a state of complete confounding.

**Unbiased Reward Model Loss.** However, during the reinforcement learning phase, we utilize only the prompts $x$ from the dataset. Since the static preference pairs $(y_w, y_l)$ are absent during this stage, we cannot directly apply Theorem 4.2 to recover the unbiased reward signal $r_\phi^*(x, y)$ for policy guidance. To address this, we require a robust loss function that trains the unbiased reward model $r_\phi^*$ directly from the noisy preference dataset. By leveraging Theorem 4.2, we can establish a mathematical relationship between $r^\eta$ and $r^*$. Given that the normal RM loss $\mathcal{L}_{\text{RM}}$

optimizes the noisy reward model $r^\eta$, substituting $r^\eta$ with its representation in terms of the unbiased $r^*$ allows us to optimize $r^*$ directly. The unbiased reward model loss is derived as follows:

**Corollary 4.3.** *Unbiased Reward Model (URM) loss:*

$$\mathcal{L}_{URM} = -\log \frac{\exp\big(r_\phi(x, y_w) - r_\phi(x, y_l)\big) + a}{\exp\big(r_\phi(x, y_w) - r_\phi(x, y_l)\big) + 1}. \quad (10)$$

*Using the URM loss, we can directly train the unbiased reward model $r_\phi^*$ from the noisy preference $\mathcal{P}^\eta$.*

Mathematically, $a$ is defined as $\frac{\eta}{1-\eta}$. However, because the true value of $\eta$ is unobservable in practice, we treat $a$ as a tunable hyperparameter during training. By adjusting the parameter $a$, we can directly train an unbiased reward model. Then, during the second stage of RL training, the interference from the noisy preference will no longer occur. Our method is orthogonal to specific RL algorithms and can be combined with any RL algorithm, such as PPO (Schulman et al., 2017) and GRPO (Shao et al., 2024). The specific RL algorithms are not the focus of this paper.

### 4.2. Unbiased Direct Preference Optimization

DPO directly supervises policy model training using preference data, so noisy preferences can significantly impact model performance. Similar to the discussion of unbiased reward model learning, we first obtain the optimal solution for normal DPO in the presence of noisy preferences.

**Lemma 4.4.** *Under noisy preferences, the normal DPO loss* $-\mathbb{E}_{(x,y_w,y_l)\sim\mathcal{P}^\eta} \log \sigma \left( \beta \log \frac{\pi_\theta(y_w|x)\pi_{ref}(y_l|x)}{\pi_\theta(y_l|x)\pi_{ref}(y_w|x)} \right)$ *is equivalent to the following noisy optimization loss:*

$$\mathbb{E}_{(x,y_w,y_l)\sim\mathcal{P}^*} \log \frac{\sigma \left( \beta \log \frac{\pi_\theta(y_w|x)\pi_{ref}(y_l|x)}{\pi_\theta(y_l|x)\pi_{ref}(y_w|x)} \right)^{\eta-1}}{\sigma \left( \beta \log \frac{\pi_\theta(y_l|x)\pi_{ref}(y_w|x)}{\pi_\theta(y_w|x)\pi_{ref}(y_l|x)} \right)^{\eta}}. \quad (11)$$

*The corresponding optimal noisy policy model satisfies:*

$$\sigma \left( \beta \log \frac{\pi_\theta^\eta(y_w|x)\pi_{ref}(y_l|x)}{\pi_\theta^\eta(y_l|x)\pi_{ref}(y_w|x)} \right) = $$
$$(1 - \eta)\, p^*(y_w \succ y_l|x) + \eta\, p^*(y_l \succ y_w|x). \quad (12)$$

**Noisy DPO Model Correction.** Based on Lemma 4.4, the noisy policy model $\pi_\theta^\eta$ obtained through DPO under noisy preferences can be corrected in the following:

**Theorem 4.5.** *For any $(x, y_w, y_l) \sim \mathcal{P}^*$, if $\pi_\theta^\eta(y|x)$ is the optimal noisy policy model, then the optimal unbiased policy model $\pi_\theta^*(y|x)$ satisfies:*

$$\left( \frac{\pi_\theta^*(y_w|x)\, \pi_{ref}(y_l|x)}{\pi_\theta^*(y_l|x)\, \pi_{ref}(y_w|x)} \right)^\beta = \frac{\left( \frac{\pi_\theta^\eta(y_w|x)\, \pi_{ref}(y_l|x)}{\pi_\theta^\eta(y_l|x)\, \pi_{ref}(y_w|x)} \right)^\beta - a}{1 - a \cdot \left( \frac{\pi_\theta^\eta(y_w|x)\, \pi_{ref}(y_l|x)}{\pi_\theta^\eta(y_l|x)\, \pi_{ref}(y_w|x)} \right)^\beta}. \quad (13)$$

*where $a = \frac{\eta}{1-\eta}$ is a constant.*

Consequently, the unbiased policy $\pi_\theta^*(y|x)$ can be recovered from the noisy policy $\pi_\theta^\eta(y|x)$. Notably, when $a = 0$, the policy ratios coincide: $\frac{\pi_\theta^*(y_w|x)\,\pi_{ref}(y_l|x)}{\pi_\theta^*(y_l|x)\,\pi_{ref}(y_w|x)} = \frac{\pi_\theta^\eta(y_w|x)\,\pi_{ref}(y_l|x)}{\pi_\theta^\eta(y_l|x)\,\pi_{ref}(y_w|x)}$, indicating that the noisy policy is unconfounded. In contrast, as $a \to \infty$, the relationship flips: $\frac{\pi_\theta^*(y_w|x)\,\pi_{ref}(y_l|x)}{\pi_\theta^*(y_l|x)\,\pi_{ref}(y_w|x)} = \frac{\pi_\theta^\eta(y_l|x)\,\pi_{ref}(y_w|x)}{\pi_\theta^\eta(y_w|x)\,\pi_{ref}(y_l|x)}$, corresponding to complete confounding, where the noisy policy effectively inverts the true human preferences.

**Unbiased DPO Loss.** In practical training, ground-truth labels are often unavailable, so we cannot determine which response is truly preferred in a noisy dataset. As a result, Theorem 4.5 cannot be applied directly to policy optimization. Motivated by Theorem 4.5, we instead establish a relationship between the noisy policy $\pi_\theta^\eta$ and the unbiased policy $\pi_\theta^*$. Specifically, we can express $\pi_\theta^\eta$ in terms of $\pi_\theta^*$ (the desired unbiased model) and substitute this equivalent into the objective. This yields an unbiased loss function for optimizing $\pi_\theta^*$ under noisy preferences as follows:

**Corollary 4.6.** *Unbiased Direct Preference Optimization (UDPO) loss:*

$$\mathcal{L}_{UDPO} = -\log \frac{\left( \frac{\pi_\theta(y_w|x)\pi_{ref}(y_l|x)}{\pi_\theta(y_l|x)\pi_{ref}(y_w|x)} \right)^\beta + a}{\left( \frac{\pi_\theta(y_w|x)\pi_{ref}(y_l|x)}{\pi_\theta(y_l|x)\pi_{ref}(y_w|x)} \right)^\beta + 1}. \quad (14)$$

*Using the UDPO loss, we can directly train the unbiased policy model $\pi_\theta^*$ from the noisy preference $\mathcal{P}^\eta$.*

By adjusting the parameter $a$, we can obtain an unbiased policy model using supervised preference training.

### 4.3. Further Theoretical Analysis

In this subsection, we further analyze the properties of the proposed URM and UDPO losses.

**A Unified Framework.** Although the URM and UDPO losses are derived independently from the normal RM and DPO losses, respectively, our resulting formulations show that they still preserve the fundamental transformation between explicit and implicit reward models in DPO and RLHF (Rafailov et al., 2023): $r_\phi(x, y_w) - r_\phi(x, y_l) = \beta \log \frac{\pi_\theta(y_w|x)}{\pi_{ref}(y_w|x)} - \beta \log \frac{\pi_\theta(y_l|x)}{\pi_{ref}(y_l|x)}$. For notational simplicity, we define the reward margin as $\Delta = r_\phi(x, y_w) - r_\phi(x, y_l) = \beta \log \frac{\pi_\theta(y_w|x)}{\pi_{ref}(y_w|x)} - \beta \log \frac{\pi_\theta(y_l|x)}{\pi_{ref}(y_l|x)}$. This shared representation enables us to analyze both the URM and UDPO losses within a unified framework by using the unbiased loss:

$$\mathcal{L}_{\text{unbiased}}(\Delta) = -\log \frac{\exp(\Delta) + a}{\exp(\Delta) + 1}. \quad (15)$$

**Gradient Analysis.** For the normal RM loss $\mathcal{L}_{\text{RM}} = -\log \sigma\left(r_\phi(x, y_w) - r_\phi(x, y_l)\right)$ and the normal DPO loss $\mathcal{L}_{\text{DPO}} = -\log \sigma\left(\beta \log \frac{\pi_\theta(y_w|x)}{\pi_{\text{ref}}(y_w|x)} - \beta \log \frac{\pi_\theta(y_l|x)}{\pi_{\text{ref}}(y_l|x)}\right)$, we can also simplify them to $\mathcal{L}(\Delta) = -\log \sigma(\Delta)$. We have the gradient as follows:

$$\frac{\partial \mathcal{L}(\Delta)}{\partial \Delta} = \sigma(\Delta) - 1. \qquad (16)$$

The gradient magnitude $\left|\frac{\partial \mathcal{L}(\Delta)}{\partial \Delta}\right|$ approaches 0 as $\Delta \gg 0$ and approaches 1 as $\Delta \ll 0$. This means that samples that are "highly misranked" (i.e., with large negative $\Delta$) receive the largest optimization weight. However, noisy or mislabeled pairs often fall into this regime (Wei et al., 2023), making the model prone to overfitting spurious or inaccurate preferences.

For the URM loss and the UDPO loss $\mathcal{L}_{\text{unbiased}}(\Delta) = -\log \frac{\exp(\Delta)+a}{\exp(\Delta)+1}$, we have the gradient as follows:

$$\frac{\partial \mathcal{L}_{\text{unbiased}}(\Delta)}{\partial \Delta} = \frac{\exp(\Delta)}{\exp(\Delta)+1} - \frac{\exp(\Delta)}{\exp(\Delta)+a} \qquad (17)$$

Notably, the gradient magnitude $|\frac{\partial \mathcal{L}_{\text{unbiased}}(\Delta)}{\partial \Delta}|$ vanishes as $\Delta \ll 0$, allowing the optimization process to effectively avoid fitting to noisy samples. Similarly, as $\Delta \gg 0$, the gradient approaches 0 in a manner consistent with the normal loss function. The gradient reaches its maximum value of $\frac{1-\sqrt{a}}{1+\sqrt{a}}$ at the intermediate point $\Delta = \frac{1}{2} \log a$. This indicates that, during optimization, the unbiased loss primarily focuses on pairs that are difficult to distinguish.

**Parameter downward Compatibility.** In practical training, we set $a = \frac{\hat{\eta}}{1-\hat{\eta}}$, with $0 \leq a < 1$. Our derivations show that the proposed loss is inherently robust when the true noise rate $\eta^*$ matches the estimate $\hat{\eta}$. Moreover, we prove that the loss remains an unbiased optimal solution when the data are cleaner than assumed, i.e., when $\eta^* \leq \hat{\eta}$.

**Corollary 4.7.** *Let $a = \frac{\hat{\eta}}{1-\hat{\eta}}$, where $0 \leq a < 1$. If the true noise rate satisfies $0 \leq \eta^* \leq \hat{\eta}$, then the unbiased loss function $\mathcal{L}_{\text{unbiased}}$ attains an unbiased optimal solution.*

In practice, the exact noise rate of a dataset is often unknown. This downward compatibility property highlights the practical utility of our method: as long as $a$ is set high to cover the potential noise, the model achieves strong results. In contrast, existing methods such as label smoothing (Mitchell, 2023) and rDPO (Chowdhury et al., 2024) lack this property; an inappropriate parameter choice in those methods can lead to performance that is inferior to standard RM or DPO loss.

**Classification Calibration and Excess Risk Bound.** The alignment task can be formulated as a specific instance of binary classification. Let $Y \in \{-1, +1\}$ denote the preference label. Here, $Y = +1$ indicates $(y_1 \succ y_2|x)$, while $Y = -1$ indicates $(y_1 \prec y_2|x)$. Given an input $X = (x, y_1, y_2)$, the model outputs the score difference $f(X) = \Delta = r(x, y_1) - r(x, y_2)$. The classification margin is $u = Yf(X)$. The margin-based form of the unbiased loss for classification is $\mathcal{L}_{\text{unbiased}}(u) = -\log \frac{\exp(u)+a}{\exp(u)+1}$.

Classification calibration (Bartlett et al., 2006), also known as Fisher consistency (Lin, 2004), is an important property ensuring that minimizing a surrogate loss can obtain the Bayes-optimal classifier in binary classification.

**Definition 4.8.** A loss function $\mathcal{L}$ is classification-calibrated if the classifier which minimizes this surrogate loss function is identical to the Bayes optimal classifier that minimizes the 0-1 loss (classification error).

We demonstrate that our proposed loss satisfies this condition.

**Theorem 4.9.** *Unbiased loss function $\mathcal{L}_{\text{unbiased}}$ is classification-calibrated.*

Based on classification calibration, we establish a relationship between the excess risks with respect to the 0–1 loss and with respect to our unbiased loss. In particular, let $R(f)$ denote the risk based on 0–1 loss, and let $R^* = \inf_f R(f)$ denote the Bayes risk. Similarly, let $R_{\mathcal{L}}(f) = \mathbb{E}_{(X,Y)}\mathcal{L}(Yf(X))$ be called the $\mathcal{L}$-risk and let $R_{\mathcal{L}}^* = \inf_f R_{\mathcal{L}}(f)$ denote the optimal $\mathcal{L}$-risk.

**Corollary 4.10.** *The 0-1 loss excess risk $R(f) - R^*$ and the $\mathcal{L}_{\text{unbiased}}$ excess risk $R_{\mathcal{L}_{\text{unbiased}}}(f) - R_{\mathcal{L}_{\text{unbiased}}}^*$ satisfy:*

$$\psi(R(f) - R^*) \leq R_{\mathcal{L}_{\text{unbiased}}}(f) - R_{\mathcal{L}_{\text{unbiased}}}^*, \qquad (18)$$

*where $\psi : [0,1] \to [0,\infty)$ is a piecewise convex function:*

$$\psi(\rho) = \begin{cases} \log 2 - \mathcal{H}_{bin}\left(\frac{1+\rho}{2}\right) & \text{if } 0 \leq \rho < \frac{1-a}{1+a} \\ \log \frac{2}{1+a} + \frac{1-\rho}{2} \log a & \text{if } \frac{1-a}{1+a} \leq \rho \leq 1 \end{cases}, \qquad (19)$$

*and $\mathcal{H}_{bin}(p) = -p \log p - (1-p) \log(1-p)$ is the binary entropy function.*

This bound implies classification consistency: if a hypothesis $f$ achieves the optimal surrogate risk, i.e., $R_{\mathcal{L}_{\text{unbiased}}}(f) = R_{\mathcal{L}_{\text{unbiased}}}^*$, then it must also achieve the Bayes-optimal classification risk, i.e., $R(f) = R^*$. Furthermore, we analyze the behavior of $\psi(\rho)$. For $\mathcal{H}_{\text{bin}}$, we perform a Taylor expansion around $p = \frac{1}{2}$: $\mathcal{H}_{\text{bin}}(p) \approx \mathcal{H}_{\text{bin}}(\frac{1}{2}) + \mathcal{H}'_{\text{bin}}(\frac{1}{2})(p - \frac{1}{2}) + \frac{1}{2}\mathcal{H}''_{\text{bin}}(\frac{1}{2})(p - \frac{1}{2})^2 = \log 2 - 2(p - \frac{1}{2})^2$. Substituting $p = \frac{1+\rho}{2}$, we obtain $\mathcal{H}_{\text{bin}}\left(\frac{1+\rho}{2}\right) \approx \log 2 - \frac{\rho^2}{2}$. Consequently, in the interval $0 \leq \rho < \frac{1-a}{1+a}$, $\psi(\rho) \approx \frac{\rho^2}{2}$, similar to Logistic loss or Quadratic loss. Conversely, in the interval $\frac{1-a}{1+a} \leq \rho \leq 1$, $\psi(\rho)$ is linear with respect to $\rho$, similar to Hinge loss. This hybrid behavior suggests that the unbiased loss inherits desirable properties from both regimes: a

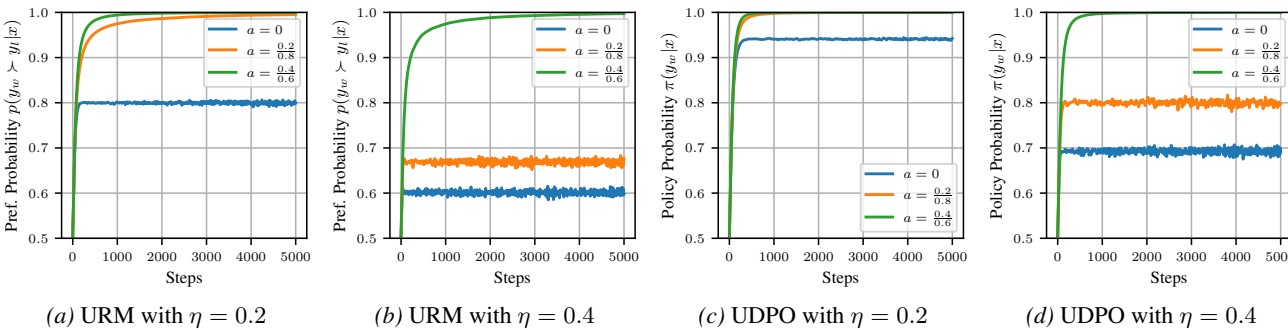

*Figure 2.* (a) & (b): Trained preference probability $p(y_w \succ y_l|x)$ for reward model using URM loss with $\eta \in \{0.2, 0.4\}$. (c) & (d): Trained policy probability $\pi(y_w|x)$ for policy model using UDPO loss with $\beta = 0.5$ and $\eta \in \{0.2, 0.4\}$.

quadratic or Logistic-like region that provides smooth, fine-grained optimization near the optimum, and a hinge-like linear region that can be more robust to high-error (potentially noisy or outlier) samples. As a result, the loss can mitigate the impact of noisy data while preserving sufficient fitting ability.

**Property Visualization.** We conduct a synthetic experiment to validate our theories and illustrate the properties of our method. Specifically, we sample preferences from one triplet $(x, y_w, y_l)$ with noise rate $\eta \in \{0.2, 0.4\}$. We train a reward model $r_\phi$ using URM loss, setting $a$ to 0 (normal RM loss), $\frac{0.2}{0.8}$ (corresponding $\eta = 0.2$), and $\frac{0.4}{0.6}$ (corresponding $\eta = 0.4$), respectively. Similarly, we apply the UDPO loss to train a policy model $\pi_\theta$ under these same values of $a$, while fixing $\beta = 0.5$ and employing a uniform distribution for $\pi_{\text{ref}}$. The models are trained for 5000 steps using the Adam optimizer with a learning rate of 0.01 and a batch size of 4096. The results are visualized in Figure 2.

It can be seen that the preference probability $p(y_w \succ y_l|x) = \sigma(r_\phi(x, y_w) - r_\phi(x, y_l))$ obtained via the normal RM loss is significantly biased by the noise rate, converging toward the value of $1 - \eta$. In contrast, our URM loss demonstrates both noise-tolerance and parameter downward compatibility. Specifically, when $a = \frac{0.2}{0.8}$, the model remains robust to the 0.2 noise rate; when $a = \frac{0.4}{0.6}$, it maintains robustness across both 0.2 and 0.4 noise rates. For the UDPO loss, the preference probability $p(y_w \succ y_l|x) = \sigma\left(\beta \log \frac{\pi_\theta(y_w|x)\pi_{\text{ref}}(y_l|x)}{\pi_\theta(y_l|x)\pi_{\text{ref}}(y_w|x)}\right)$ exhibits behavior identical to that of the URM loss. To further illustrate the specific characteristics of the UDPO loss, we report the policy probability $\pi_\theta(y_w|x)$, which likewise exhibits noise-tolerance and parameter downward compatibility. These visualizations remain consistent with our theoretical analysis.

**Normalized Unbiased Loss.** Gradient analysis reveals that the maximum gradient magnitude of the unbiased loss, $\max_\Delta \left| \frac{\partial \mathcal{L}_{\text{unbiased}}(\Delta)}{\partial \Delta} \right| = \frac{1-\sqrt{a}}{1+\sqrt{a}}$, decreases as $a$ increases. This reduction may imply that, relative to the normal loss, optimizing the unbiased loss may require a larger learning

rate to achieve comparable update magnitudes. To avoid additional learning rate adjustments, we can normalize the unbiased loss by a constant $\alpha = \frac{1+\sqrt{a}}{1-\sqrt{a}}$. With this scaling, the maximum gradient magnitude is preserved as $\max_\Delta \left| \frac{\partial \mathcal{L}_{\text{unbiased}}(\Delta)}{\partial \Delta} \right| = 1$. We refer to the normalized variants as $\alpha$-URM and $\alpha$-UDPO. Since this modification is merely a constant rescaling of the objective, it does not alter the theoretical properties of URM and UDPO.

**Instance-Dependent Noise.** Our theory can naturally extend to instance-dependent noise. Specifically, let $\eta_x$ denote the noise rate for instance $x$, and define the corresponding constant $a_x$ as $\frac{\eta_x}{1-\eta_x}$. Since the proofs of the noise correction process in Theorem 4.2 and Theorem 4.5 are derived pointwise, the instance-dependent versions follow directly by replacing the constant $a$ with $a_x$. Although the $\eta_x$ of each instance is unknown, our parameter downward compatibility further implies that it suffices to use a single global parameter $\hat{a}$ satisfying $\hat{a} \geq \sup_x a_x$ (equivalently, $\hat{\eta} \geq \sup_x \eta_x$). In this case, the unbiased loss still recovers the unbiased optimal solution.

## 5. Experiments

In this section, we validate the effectiveness of our proposed methods through comprehensive experiments, spanning reward model training and supervised preference training. Detailed experimental settings are provided in the Appendix B.

**Datasets.** We evaluate our methods on three widely used real-world datasets, including the dialogue dataset Anthropic-HH (helpful-base) (Bai et al., 2022), the summarization dataset Reddit TL;DR (Völske et al., 2017), and the comprehensive dataset UltraFeedback Binarized (UFB) (Cui et al., 2023).

**Baselines.** We compare our method with several state-of-the-art approaches, with a particular focus on robustness-oriented methods. For reward model training, we consider the standard RM (Christiano et al., 2017), label smoothing (cRM), and rRM (Chowdhury et al., 2024). For supervised

*Table 1.* Win rates (%) of different reward model learning methods vs. the SFT model under different manual flip rates (0%, 20%, 40%). The top-2 best results are in bold.

| Models | Methods | HH | | | TL;DR | | | Average |
|---|---|---|---|---|---|---|---|---|
| | | 0% | 20% | 40% | 0% | 20% | 40% | |
| Llama-3.2-3B | RM | 80.0 | 78.3 | 69.7 | 79.4 | 76.4 | 69.1 | 75.5 |
| | cRM | 80.7 | 76.1 | 70.1 | 80.8 | 76.5 | 69.9 | 75.7 |
| | rRM | 79.9 | 78.3 | 72.4 | 77.3 | 77.7 | 68.6 | 75.7 |
| | **URM** | **81.4** | **78.9** | **72.6** | **82.8** | **83.2** | **77.5** | **79.4** |
| | $\alpha$-**URM** | **82.5** | **81.4** | **74.3** | **83.4** | **81.9** | **76.1** | **79.9** |
| Qwen-3-1.7B | RM | 63.4 | 55.5 | 43.9 | 81.3 | 76.2 | 63.3 | 63.9 |
| | cRM | 64.2 | 56.0 | 46.2 | 80.4 | 76.6 | 66.4 | 65.0 |
| | rRM | 65.7 | **62.5** | 52.1 | 76.8 | 73.6 | 64.7 | 65.9 |
| | **URM** | **67.6** | 61.8 | **58.2** | **81.5** | **80.0** | **71.6** | **70.1** |
| | $\alpha$-**URM** | **66.9** | 61.6 | **57.9** | **82.9** | **79.2** | **68.9** | **69.6** |

*Table 2.* Win rates (%) of different supervised preference learning methods vs the SFT model under different manual flip rates (0%, 20%, 40%). The top-2 best results are in bold.

| Models | Methods | HH | | | TL;DR | | | Average |
|---|---|---|---|---|---|---|---|---|
| | | 0% | 20% | 40% | 0% | 20% | 40% | |
| Llama-3.2-3B | DPO | 85.4 | 80.4 | 69.8 | 72.2 | 64.6 | 56.1 | 71.4 |
| | cDPO | 85.3 | 80.2 | 70.9 | 69.6 | 61.0 | 57.0 | 70.7 |
| | IPO | 85.6 | 89.3 | 80.0 | 86.5 | 80.3 | 69.3 | 81.8 |
| | rDPO | 88.0 | 89.3 | 84.4 | 85.9 | 83.5 | 77.3 | 84.7 |
| | Dr.DPO | 88.2 | 88.2 | 81.6 | **86.8** | 84.3 | **82.9** | 85.0 |
| | **UDPO** | **90.8** | **90.4** | **85.6** | 86.3 | **85.3** | 77.2 | **85.9** |
| | $\alpha$-**UDPO** | **90.4** | **89.8** | **85.7** | **87.2** | **86.4** | 78.2 | **86.1** |
| Qwen-3-1.7B | DPO | 75.4 | 68.2 | 58.2 | 73.9 | 65.4 | 54.5 | 65.9 |
| | cDPO | 72.8 | 70.1 | 59.3 | 67.2 | 60.5 | 51.9 | 63.6 |
| | IPO | 71.7 | 68.7 | 63.3 | 82.1 | 78.9 | 67.1 | 72.0 |
| | rDPO | 78.8 | 72.9 | 68.1 | 84.3 | 72.4 | 70.2 | 74.5 |
| | Dr.DPO | 80.2 | 77.7 | 68.2 | 82.9 | 80.1 | 70.5 | 76.6 |
| | **UDPO** | **83.4** | **81.2** | **73.0** | 84.6 | **83.0** | 71.3 | **79.4** |
| | $\alpha$-**UDPO** | **81.3** | **80.3** | **73.2** | **86.3** | **84.9** | 71.5 | **79.6** |

preference training, we consider DPO (Rafailov et al., 2023), label smoothing (cDPO), IPO (Azar et al., 2024), rDPO (Chowdhury et al., 2024), and Dr.DPO (Wu et al., 2025). We search for the optimal hyperparameter for each baseline to ensure a fair comparison.

### 5.1. Evaluations on HH and TL;DR

We first conduct evaluations on the dialogue dataset HH and the summarization dataset TL;DR.

**Setup.** We use the Llama-3.2-3B (Grattafiori et al., 2024) and Qwen-3-1.7B (Yang et al., 2025) as the base models. To obtain the SFT model, we fine-tune the base model exclu-

sively on preferred completions similar to (Rafailov et al., 2023). For reward model evaluation, following (Nakano et al., 2021; Gao et al., 2023), we employ a best-of-$n$ sampling strategy. Specifically, we let the SFT model generate $n = 20$ responses, and then use the reward model to select the highest-scoring response, which is compared with the default output of the SFT model. The best-of-$n$ metric offers a more stable comparison of reward model quality. For supervised preference training evaluation, following (Rafailov et al., 2023), we compare the responses generated by the trained policy model against those of the SFT model. We employ GPT-5 (GPT-5-chat-latest) as the judge to calculate the win rate. Beyond the inherent noise in the HH and TL;DR datasets, we also investigate scenarios with 20% and

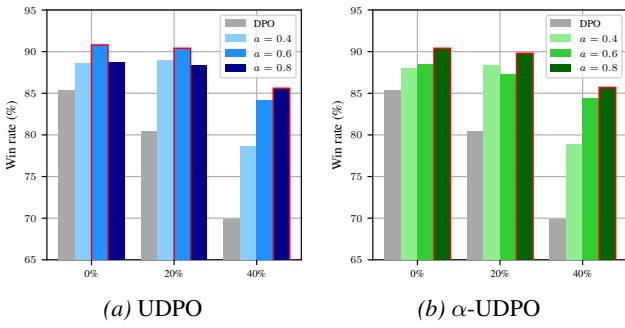

*Figure 3.* The ablation results of UDPO and $\alpha$-UDPO using Llama-3.2-3B on HH with different manual flip rates (0%, 20%, 40%). The best results in each case are highlighted with a red border.

*Table 3.* Win rates (%) of different reward model learning methods vs. the SFT model on UFB. The top-2 best results are in bold.

| Methods | UFB | |
|---|---|---|
| | Llama-3.1-8B | Qwen-3-8B |
| RM | 68.9 | 54.3 |
| cRM | 66.5 | 52.9 |
| rRM | 63.6 | 63.9 |
| **URM** | **72.5** | **64.6** |
| $\alpha$**-URM** | **72.3** | **64.9** |

*Table 4.* Win rates (%) of different supervised preference learning methods vs the SFT model on UFB. The top-2 best results are in bold.

| Methods | UFB | |
|---|---|---|
| | Llama-3.1-8B | Qwen-3-8B |
| DPO | 65.4 | 56.3 |
| cDPO | 63.1 | 55.1 |
| IPO | 72.7 | 57.8 |
| rDPO | 72.1 | 58.3 |
| Dr.DPO | 65.8 | 56.5 |
| **UDPO** | **74.3** | **60.6** |
| $\alpha$**-UDPO** | **73.0** | **60.7** |

40% manually flipped labels to simulate higher noise levels. Please note that the original dataset already contains noise; therefore, a manual flip rate of 0% does not imply no noise.

**Results.** We report the win rates for reward model training and supervised preference learning in Table 1 and Table 2, respectively. For reward model learning (Table 1), our proposed URM and $\alpha$-URM losses consistently outperform all baselines in the original setting (0% flip rate), indicating that they effectively mitigate intrinsic noise in real-world data. As we inject additional synthetic noise, our methods remain more robust. For example, with Llama-3.2-3B on TL;DR at a 40% flip rate, URM and $\alpha$-URM improve win rates by approximately 8% over the baselines. Relative to the normal RM loss, cRM and rRM do not yield a meaningful improvement in average win rate. In contrast, across both Llama-3.2-3B and Qwen-3-1.7B, our methods exceed the SOTA baseline by roughly 4% in average win rate. For supervised preference learning (Table 2), our methods have achieved the best results in most cases. The average win rate of UDPO and $\alpha$-UDPO exceeded that of normal DPO by around 15%. This substantial gap confirms that normal DPO is prone to overfitting noisy preferences, while our unbiased framework successfully recovers the unbiased policy. UDPO and $\alpha$-UDPO consistently achieve the highest average win rates across all datasets, which further validates their effectiveness in both naturally noisy and highly corrupted environments.

**Ablation Experiment.** It can be observed that the unbiased loss and the normalized unbiased loss achieve comparable strong performance. We further report the hyperparameter ablation results using UDPO and $\alpha$-UDPO of Llama-3.2-3B on the HH dataset, as shown in Figure 3. The results indicate that UDPO is more sensitive to hyperparameter choices, whereas the normalized $\alpha$-UDPO consistently attains optimal performance across settings when using a larger $a$. This behavior is consistent with our parameter downward compatibility theory.

## 5.2. Evaluations on UFB.

We conduct experiments using larger models on the more complex dataset UFB.

**Setup.** We use the Llama-3.1-8B (Grattafiori et al., 2024) and Qwen-3-8B (Yang et al., 2025) as the base models. To obtain the SFT model, we fine-tune the base model on the instruct dataset Capybara (Daniele & Deeprasit, 2023). The assessment method is consistent with HH and TL;DR.

**Results.** We report the win rates for reward model training and supervised preference training in Table 3 and Table 4, respectively. For reward model learning (Table 3), our proposed URM and $\alpha$-URM losses consistently outperform all baselines. Notably, on the Qwen-3-8B model, while the normal RM performs poorly with a 54.3% win rate, our URM and $\alpha$-URM achieve significantly higher win rates of 64.6% and 64.9% respectively, surpassing the normal RM loss. For supervised preference learning (Table 4), UDPO and $\alpha$-UDPO consistently achieve the best results. Our methods outperform the SOTA methods by approximately 2% on both models. These results on the complex UFB dataset with larger models further validate the scalability and robustness of our unbiased alignment framework.

# 6. Conclusion

This work introduces an unbiased alignment framework for LLM alignment with noisy preferences. By modeling preference corruption as a mathematical transition process and analytically inverting the distortion, we derive two practical objectives: the *Unbiased Reward Model* (URM) loss for reward learning in RLHF, and the *Unbiased Direct Preference Optimization* (UDPO) loss for supervised preference optimization. We further show that both objectives admit a unified margin-based form that preserves the fundamental RLHF-DPO equivalence, constituting a unified framework. Our theoretical analysis establishes noise-tolerance, parameter downward compatibility, and classification calibration. Extensive experiments demonstrate that our methods show consistent improvements over strong baselines. Overall, our work offers a concise, interpretable, and effective route to robust LLM alignment.

## Impact Statement

This work presents a novel advance in LLM robust alignment, with positive implications for developing safe and reliable AI systems. This work is not expected to have a negative social impact.

## Acknowledgement

This work was supported by National Natural Science Foundation of China under Grants 62525107.

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

# A. Proofs

### Proof of Lemma 4.1

*Proof.* Let $\hat{p}(y_w \succ y_l|x) = \sigma\big(r_\phi(x, y_w) - r_\phi(x, y_l)\big)$, under the noisy preference, we have:

$$
\begin{aligned}
\mathcal{L}_{\text{RM}} &= -\big[(1-\eta)\log\sigma\big(r_\phi(x, y_w) - r_\phi(x, y_l)\big) + \eta\log\sigma\big(r_\phi(x, y_l) - r_\phi(x, y_w)\big)\big] \\
&= -\big[(1-\eta)\log\hat{p}(y_w \succ y_l|x) + \eta\log\hat{p}(y_l \succ y_w|x)\big].
\end{aligned}
\tag{20}
$$

Hence, for the gradient $\frac{\nabla\mathcal{L}}{\nabla\hat{p}(y_w \succ y_l|x)}$, we have:

$$
\frac{\nabla\mathcal{L}}{\nabla\hat{p}(y_w \succ y_l|x)} = -(1-\eta)\frac{1}{\hat{p}(y_w \succ y_l|x)} + \eta\frac{1}{1 - \hat{p}(y_w \succ y_l|x)}.
\tag{21}
$$

Let $\frac{\nabla\mathcal{L}}{\nabla\hat{p}(y_w \succ y_l|x)} = 0$, because $p^*(y_w \succ y_l|x) = 1$ and $p^*(y_l \succ y_w|x) = 0$, we have the optimal solution $\hat{p}(y_w \succ y_l|x) = 1 - \eta = (1-\eta)p^*(y_w \succ y_l|x) + \eta p^*(y_l \succ y_w|x)$. □

### Proof of Theorem 4.2

*Proof.* Referencing Lemma 4.1, we know the relationship between the noisy preference probability and the unbiased preference probability is:

$$
\begin{aligned}
\sigma(r_\phi^\eta(x, y_w) - r_\phi^\eta(x, y_l)) &= (1-\eta)\cdot\sigma(r_\phi^*(x, y_w) - r_\phi^*(x, y_l)) + \eta\cdot\sigma(r_\phi^*(x, y_l) - r_\phi^*(x, y_w)) \\
&= (1-\eta)\cdot\sigma(r_\phi^*(x, y_w) - r_\phi^*(x, y_l)) + \eta\cdot(1 - \sigma(r_\phi^*(x, y_w) - r_\phi^*(x, y_l))) \\
&= (1-2\eta)\cdot\sigma(r_\phi^*(x, y_w) - r_\phi^*(x, y_l)) + \eta
\end{aligned}
\tag{22}
$$

To simplify the symbols, we set $\Delta^\eta = r_\phi^\eta(x, y_w) - r_\phi^\eta(x, y_l)$ and $\Delta^* = r_\phi^*(x, y_w) - r_\phi^*(x, y_l)$. Opening the sigmoid function $\sigma$, we have:

$$
\begin{aligned}
&\frac{\exp(r_\phi^\eta(x, y_w))}{\exp(r_\phi^\eta(x, y_w)) + \exp(r_\phi^\eta(x, y_l))} = (1-2\eta)\cdot\frac{\exp(r_\phi^*(x, y_w))}{\exp(r_\phi^*(x, y_w)) + \exp(r_\phi^*(x, y_l))} + \eta \\
&\Rightarrow \frac{\exp(\Delta^\eta)}{\exp(\Delta^\eta) + 1} = (1-2\eta)\cdot\frac{\exp(\Delta^*)}{\exp(\Delta^*) + 1} + \eta \\
&\Rightarrow \exp(\Delta^\eta)(\exp(\Delta^*) + 1) = (\exp(\Delta^\eta) + 1)[\eta + (1-\eta)\exp(\Delta^*)] \\
&\Rightarrow \exp(\Delta^*)[\exp(\Delta^\eta) - (1-\eta)\exp(\Delta^\eta) - (1-\eta)] = \eta\exp(\Delta^\eta) + \eta - \exp(\Delta^\eta) \\
&\Rightarrow \exp(\Delta^*) = \frac{\eta\exp(\Delta^\eta) + \eta - \exp(\Delta^\eta)}{\exp(\Delta^\eta) - (1-\eta)\exp(\Delta^\eta) - (1-\eta)} \\
&\Rightarrow \exp(\Delta^*) = \frac{(\eta-1)\exp(\Delta^\eta) + \eta}{\eta\exp(\Delta^\eta) - (1-\eta)} = \frac{\exp(\Delta^\eta) - \frac{\eta}{1-\eta}}{1 - \frac{\eta}{1-\eta}\exp(\Delta^\eta)} \\
&\Rightarrow \Delta^* = \log\left[\frac{\exp(\Delta^\eta) - \frac{\eta}{1-\eta}}{1 - \frac{\eta}{1-\eta}\exp(\Delta^\eta)}\right]. \\
&\Rightarrow r_\phi^*(x, y_w) - r_\phi^*(x, y_l) = \log\left[\frac{\exp\big(r_\phi^\eta(x, y_w) - r_\phi^\eta(x, y_l)\big) - a}{1 - a\cdot\exp\big(r_\phi^\eta(x, y_w) - r_\phi^\eta(x, y_l)\big)}\right],
\end{aligned}
\tag{23}
$$

where $a = \frac{\eta}{1-\eta}$ is a constant. □

### Proof of Corollary 4.3

*Proof.* Refer to Theorem 4.2, we have :

$$
\begin{aligned}
\exp(\Delta^*) &= \frac{\exp(\Delta^\eta) - a}{1 - a \cdot \exp(\Delta^\eta)} \\
\Rightarrow \exp(\Delta^*)(1 - a \cdot \exp(\Delta^\eta)) &= \exp(\Delta^\eta) - a \\
\Rightarrow \exp(\Delta^*) - a \cdot \exp(\Delta^*)\exp(\Delta^\eta) &= \exp(\Delta^\eta) - a \\
\Rightarrow \exp(\Delta^\eta) + a \cdot \exp(\Delta^*)\exp(\Delta^\eta) &= \exp(\Delta^*) + a \\
\Rightarrow \exp(\Delta^\eta) &= \frac{\exp(\Delta^*) + a}{1 + a \cdot \exp(\Delta^*)} \\
\Rightarrow \Delta^\eta &= \log\Big[\frac{\exp(\Delta^*) + a}{1 + a \cdot \exp(\Delta^*)}\Big]. \\
\Rightarrow r_\phi^\eta(x, y_w) - r_\phi^\eta(x, y_l) &= \log\Big[\frac{\exp(r_\phi^*(x, y_w) - r_\phi^*(x, y_l)) + a}{1 + a \cdot \exp(r_\phi^*(x, y_w) - r_\phi^*(x, y_l))}\Big]
\end{aligned}
\tag{24}
$$

Under the noise preference, the normal reward model loss $\mathcal{L}_{\mathrm{RM}}$ optimize the noisy reward model $r_\phi^\eta$, i.e. $\mathcal{L}_{\mathrm{RM}} = -\log \sigma(r_\phi^\eta(x, y_w) - r_\phi^\eta(x, y_l))$. Therefore, by replacing $r_\phi^\eta(x, y_w) - r_\phi^\eta(x, y_l)$ as $\log\Big[\frac{\exp(r_\phi^*(x,y_w)-r_\phi^*(x,y_l))+a}{1+a\cdot\exp(r_\phi^*(x,y_w)-r_\phi^*(x,y_l))}\Big]$, we can directly optimize the unbiased reward model $r_\phi^*$. Our unbiased reward model loss as follows:

$$
\begin{aligned}
\mathcal{L}_{\mathrm{URM}} &= -\log \sigma\Big(\log\Big[\frac{\exp(r_\phi^*(x, y_w) - r_\phi^*(x, y_l)) + a}{1 + a \cdot \exp(r_\phi^*(x, y_w) - r_\phi^*(x, y_l))}\Big]\Big) \\
&= -\log \frac{1}{1 + \frac{1 + a\cdot\exp(\Delta^*)}{\exp(\Delta^*) + a}} \\
&= -\log \frac{\exp(\Delta^*) + a}{\exp(\Delta^*) + a + 1 + a \cdot \exp(\Delta^*)} \\
&= -\log \frac{\exp(\Delta^*) + a}{(a + 1)(\exp(\Delta^*) + 1)} \\
&= -[\log \frac{\exp(\Delta^*) + a}{\exp(\Delta^*) + 1} - \log(a + 1)].
\end{aligned}
\tag{25}
$$

Since $\log(a + 1)$ is a constant, we can disregard it. Therefore, our final loss for training is $\mathcal{L}_{\mathrm{URM}} = -\log \frac{\exp(r_\phi(x,y_w)-r_\phi(x,y_l))+a}{\exp(r_\phi(x,y_w)-r_\phi(x,y_l))+1}$. The derivation of Equation 25 enables us to perform less operations of $\sigma$ and $\log$ when calculating the loss.

$\square$

## Proof of Lemma 4.4

*Proof.* Let $\hat{p}(y_w \succ y_l|x) = \sigma\Big(\beta \log \frac{\pi_\theta(y_w|x)\pi_{\mathrm{ref}}(y_l|x)}{\pi_\theta(y_l|x)\pi_{\mathrm{ref}}(y_w|x)}\Big)$, under the noisy preference, we have:

$$
\begin{aligned}
\mathcal{L}_{\mathrm{DPO}} &= -\big[(1 - \eta)\log\sigma\Big(\beta \log \frac{\pi_\theta(y_w|x)\pi_{\mathrm{ref}}(y_l|x)}{\pi_\theta(y_l|x)\pi_{\mathrm{ref}}(y_w|x)}\Big) + \eta\log\sigma\Big(\beta \log \frac{\pi_\theta(y_l|x)\pi_{\mathrm{ref}}(y_w|x)}{\pi_\theta(y_w|x)\pi_{\mathrm{ref}}(y_l|x)}\Big)\big] \\
&= -\big[(1 - \eta)\log\hat{p}(y_w \succ y_l|x) + \eta\log\hat{p}(y_l \succ y_w|x)\big].
\end{aligned}
\tag{26}
$$

Hence, for the gradient $\frac{\nabla\mathcal{L}}{\nabla\hat{p}(y_w \succ y_l|x)}$, we have:

$$
\frac{\nabla\mathcal{L}}{\nabla\hat{p}(y_w \succ y_l|x)} = -(1 - \eta)\frac{1}{\hat{p}(y_w \succ y_l|x)} + \eta\frac{1}{1 - \hat{p}(y_w \succ y_l|x)}.
\tag{27}
$$

Let $\frac{\nabla\mathcal{L}}{\nabla\hat{p}(y_w \succ y_l|x)} = 0$, because $p^*(y_w \succ y_l|x) = 1$ and $p^*(y_l \succ y_w|x) = 0$, we have the optimal solution $\hat{p}(y_w \succ y_l|x) = 1 - \eta = (1 - \eta)p^*(y_w \succ y_l|x) + \eta p^*(y_l \succ y_w|x)$. $\square$

**Proof of Theorem 4.5**

*Proof.* Referencing Lemma 4.4, we know the relationship between the noisy preference probability and the unbiased preference probability is:

$$\sigma(\beta \log \frac{\pi_\theta^\eta(y_w|x)\pi_{\text{ref}}(y_l|x)}{\pi_\theta^\eta(y_l|x)\pi_{\text{ref}}(y_w|x)}) = (1-\eta)\sigma(\beta \log \frac{\pi_\theta^*(y_w|x)\pi_{\text{ref}}(y_l|x)}{\pi_\theta^*(y_l|x)\pi_{\text{ref}}(y_w|x)}) + \eta\sigma(\beta \log \frac{\pi_\theta^*(y_l|x)\pi_{\text{ref}}(y_w|x)}{\pi_\theta^*(y_w|x)\pi_{\text{ref}}(y_l|x)}) \tag{28}$$

To simplify the symbols, we set $U^\eta = (\frac{\pi_\theta^\eta(y_w|x)\pi_{\text{ref}}(y_l|x)}{\pi_\theta^\eta(y_l|x)\pi_{\text{ref}}(y_w|x)})^\beta$ and $U^* = (\frac{\pi_\theta^*(y_w|x)\pi_{\text{ref}}(y_l|x)}{\pi_\theta^*(y_l|x)\pi_{\text{ref}}(y_w|x)})^\beta$. Opening the sigmoid function $\sigma$, we have:

$$\frac{\exp(\beta \log \frac{\pi_\theta^\eta(y_w|x)\pi_{\text{ref}}(y_l|x)}{\pi_\theta^\eta(y_l|x)\pi_{\text{ref}}(y_w|x)})}{1+\exp(\beta \log \frac{\pi_\theta^\eta(y_w|x)\pi_{\text{ref}}(y_l|x)}{\pi_\theta^\eta(y_l|x)\pi_{\text{ref}}(y_w|x)})} = (1-\eta)\frac{\exp(\beta \log \frac{\pi_\theta^*(y_w|x)\pi_{\text{ref}}(y_l|x)}{\pi_\theta^*(y_l|x)\pi_{\text{ref}}(y_w|x)})}{1+\exp(\beta \log \frac{\pi_\theta^*(y_w|x)\pi_{\text{ref}}(y_l|x)}{\pi_\theta^*(y_l|x)\pi_{\text{ref}}(y_w|x)})} + \eta\left(1 - \frac{\exp(\beta \log \frac{\pi_\theta^*(y_w|x)\pi_{\text{ref}}(y_l|x)}{\pi_\theta^*(y_l|x)\pi_{\text{ref}}(y_w|x)})}{1+\exp(\beta \log \frac{\pi_\theta^*(y_w|x)\pi_{\text{ref}}(y_l|x)}{\pi_\theta^*(y_l|x)\pi_{\text{ref}}(y_w|x)})}\right)$$

$$\Rightarrow \frac{U^\eta}{1+U^\eta} = (1-\eta)\frac{U^*}{1+U^*} + \eta\left(1 - \frac{U^*}{1+U^*}\right)$$

$$\Rightarrow \frac{U^\eta}{1+U^\eta} = (1-\eta)\frac{U^*}{1+U^*} + \eta\frac{1}{1+U^*}$$

$$\Rightarrow \frac{U^\eta}{1+U^\eta} = \frac{(1-\eta)U^* + \eta}{1+U^*}$$

$$\Rightarrow U^\eta(1+U^*) = (1+U^\eta)((1-\eta)U^* + \eta)$$

$$\Rightarrow U^\eta + U^\eta U^* = (1-\eta)U^* + \eta + (1-\eta)U^\eta U^* + \eta U^\eta$$

$$\Rightarrow U^\eta U^* - (1-\eta)U^* - (1-\eta)U^\eta U^* = \eta + \eta U^\eta - U^\eta$$

$$\Rightarrow U^*(U^\eta - (1-\eta) - (1-\eta)U^\eta) = \eta - (1-\eta)U^\eta$$

$$\Rightarrow U^*(\eta U^\eta - 1 + \eta) = \eta - (1-\eta)U^\eta$$

$$\Rightarrow U^* = \frac{\eta - (1-\eta)U^\eta}{\eta U^\eta - (1-\eta)} = \frac{U^\eta - \frac{\eta}{1-\eta}}{1 - \frac{\eta}{1-\eta}U^\eta}$$

$$\Rightarrow (\frac{\pi_\theta^*(y_w|x)\pi_{\text{ref}}(y_l|x)}{\pi_\theta^*(y_l|x)\pi_{\text{ref}}(y_w|x)})^\beta = \frac{(\frac{\pi_\theta^\eta(y_w|x)\pi_{\text{ref}}(y_l|x)}{\pi_\theta^\eta(y_l|x)\pi_{\text{ref}}(y_w|x)})^\beta - a}{1 - a \cdot (\frac{\pi_\theta^\eta(y_w|x)\pi_{\text{ref}}(y_l|x)}{\pi_\theta^\eta(y_l|x)\pi_{\text{ref}}(y_w|x)})^\beta},$$

$$\tag{29}$$

where $a = \frac{\eta}{1-\eta}$ is a constant.

$\square$

**Proof of Corollary 4.6**

*Proof.* Refer to Theorem 4.5, we have:

$$(\frac{\pi_\theta^*(y_w|x)\pi_{\text{ref}}(y_l|x)}{\pi_\theta^*(y_l|x)\pi_{\text{ref}}(y_w|x)})^\beta = \frac{(\frac{\pi_\theta^\eta(y_w|x)\pi_{\text{ref}}(y_l|x)}{\pi_\theta^\eta(y_l|x)\pi_{\text{ref}}(y_w|x)})^\beta - a}{1 - a \cdot (\frac{\pi_\theta^\eta(y_w|x)\pi_{\text{ref}}(y_l|x)}{\pi_\theta^\eta(y_l|x)\pi_{\text{ref}}(y_w|x)})^\beta}$$

$$\Rightarrow U^*(1 - aU^\eta) = U^\eta - a$$

$$\Rightarrow U^* - aU^*U^\eta = U^* - a$$

$$\Rightarrow U^* + a = U^\eta(1 + aU^*)$$

$$\Rightarrow U^\eta = \frac{U^* + a}{1 + aU^*}$$

$$\Rightarrow (\frac{\pi_\theta^\eta(y_w|x)\pi_{\text{ref}}(y_l|x)}{\pi_\theta^\eta(y_l|x)\pi_{\text{ref}}(y_w|x)})^\beta = \frac{(\frac{\pi_\theta^*(y_w|x)\pi_{\text{ref}}(y_l|x)}{\pi_\theta^*(y_l|x)\pi_{\text{ref}}(y_w|x)})^\beta + a}{1 + a \cdot (\frac{\pi_\theta^*(y_w|x)\pi_{\text{ref}}(y_l|x)}{\pi_\theta^*(y_l|x)\pi_{\text{ref}}(y_w|x)})^\beta}$$

$$\tag{30}$$

The standard DPO loss optimizes the model likelihood against the preference dataset.

Under noisy preferences, the standard DPO loss minimizes the negative log-likelihood of the noisy probability distribution, i.e., $\mathcal{L}_{\text{DPO}} = -\log \sigma(\beta \log \frac{\pi_\theta^\eta(y_w|x)\pi_{\text{ref}}(y_l|x)}{\pi_\theta^\eta(y_l|x)\pi_{\text{ref}}(y_w|x)}) = -\log[\sigma(\log(\frac{\pi_\theta^\eta(y_w|x)\pi_{\text{ref}}(y_l|x)}{\pi_\theta^\eta(y_l|x)\pi_{\text{ref}}(y_w|x)})^\beta)]$

To obtain the unbiased loss, we substitute $(\frac{\pi_\theta^\eta(y_w|x)\pi_{\text{ref}}(y_l|x)}{\pi_\theta^\eta(y_l|x)\pi_{\text{ref}}(y_w|x)})^\beta$ with its expression in terms of $(\frac{\pi_\theta^*(y_w|x)\pi_{\text{ref}}(y_l|x)}{\pi_\theta^*(y_l|x)\pi_{\text{ref}}(y_w|x)})^\beta$ (the unbiased model we wish to train):

$$
\begin{aligned}
\mathcal{L}_{\text{UDPO}} &= -\log[\sigma(\log \frac{(\frac{\pi_\theta^*(y_w|x)\pi_{\text{ref}}(y_l|x)}{\pi_\theta^*(y_l|x)\pi_{\text{ref}}(y_w|x)})^\beta + a}{1 + a \cdot (\frac{\pi_\theta^*(y_w|x)\pi_{\text{ref}}(y_l|x)}{\pi_\theta^*(y_l|x)\pi_{\text{ref}}(y_w|x)})^\beta})] \\
&= -\log\left(\frac{\frac{U^*+a}{1+aU^*}}{1 + \frac{U^*+a}{1+aU^*}}\right) \\
&= -\log\left(\frac{U^*+a}{(1+aU^*)+(U^*+a)}\right) \\
&= -\log\left(\frac{U^*+a}{(1+a)(1+U^*)}\right) \\
&= -\log\left(\frac{U^*+a}{U^*+1}\right) + \log(1+a)
\end{aligned}
\tag{31}
$$

Since $\log(1+a)$ is a constant with respect to $\theta$, it can be discarded during optimization. Substituting the full expression for $U^* = \left(\frac{\pi_\theta(y_w|x)\pi_{\text{ref}}(y_l|x)}{\pi_\theta(y_l|x)\pi_{\text{ref}}(y_w|x)}\right)^\beta$. We arrive at the final objective function: $\mathcal{L}_{\text{UDPO}} = -\log \frac{\left(\frac{\pi_\theta(y_w|x)\pi_{\text{ref}}(y_l|x)}{\pi_\theta(y_l|x)\pi_{\text{ref}}(y_w|x)}\right)^\beta + a}{\left(\frac{\pi_\theta(y_w|x)\pi_{\text{ref}}(y_l|x)}{\pi_\theta(y_l|x)\pi_{\text{ref}}(y_w|x)}\right)^\beta + 1}$. This formula allows for training the unbiased policy $\pi_\theta^*$ directly using the noisy preference dataset. The derivation of Equation 31 enables us to perform less operations of $\sigma$ and $\log$ when calculating the loss. $\square$

**Proof of Corollary 4.7**

*Proof.* Based on Corollary 4.3 and Corollary 4.6 When $\eta^* = \hat{\eta}$, it is clearly proof. We prove the case where $\eta^* < \hat{\eta}$.

Under the noisy preference distribution, the total unbiased loss is $\mathcal{L}_{\mathcal{P}^\eta}(\Delta) = (1-\eta) * \mathcal{L}_{\text{unbiased}}(\Delta) + \eta \mathcal{L}_{\text{unbiased}}(-\Delta)$, the gradient is:

$$
\frac{\partial \mathcal{L}_{\mathcal{P}^\eta}(\Delta)}{\partial \Delta} = -(1-\eta^*)(1-a)\frac{\exp(\Delta)}{(\exp(\Delta)+a)(\exp(\Delta)+1)} + \eta^*(1-a)\frac{\exp(-\Delta)}{(\exp(-\Delta)+a)(\exp(-\Delta)+1)}.
\tag{32}
$$

We need to prove that: when $\hat{\eta} > \eta^*$, for any $\Delta$, the gradient $\frac{\partial \mathcal{L}_{\mathcal{P}^\eta}(\Delta)}{\partial \Delta}$ is always less than 0. This implies that the objective function $\mathcal{L}_{\mathcal{P}^\eta}(\Delta)$ is monotonically decreasing, and thus the optimal solution is achieved when $\Delta \to +\infty$.

The term $\frac{\exp(-\Delta)}{(\exp(-\Delta)+a)(\exp(-\Delta)+1)} = \frac{\exp(-\Delta)\exp(2\Delta)}{[(\exp(-\Delta)+a)\exp(\Delta)][(\exp(-\Delta)+1)\exp(\Delta)]} = \frac{\exp(\Delta)}{(1+a\exp(\Delta))(1+\exp(\Delta))}$. Therefore, we have:

$$
\text{Sign}\left(\frac{\partial \mathcal{L}_{\mathcal{P}^\eta}}{\partial \Delta}\right) = \text{Sign}\left(\eta^* \frac{\exp(\Delta)}{(1+a\exp(\Delta))(1+\exp(\Delta))} - (1-\eta^*)\frac{\exp(\Delta)}{(\exp(\Delta)+a)(\exp(\Delta)+1)}\right),
\tag{33}
$$

where term $(1-a) > 0$ is overlooked. Eliminating the common positive terms $\frac{\exp(\Delta)}{1+\exp(\Delta)}$, we need to prove the following inequality:

$$
\begin{aligned}
\frac{\eta^*}{1+a\exp(\Delta)} &< \frac{1-\eta^*}{\exp(\Delta)+a} \\
\Rightarrow \eta^*(\exp(\Delta)+a) &< (1-\eta^*)(1+a\exp(\Delta)) \\
\Rightarrow \eta^*\exp(\Delta)+\eta^*a &< 1-\eta^*+a\exp(\Delta)-a\eta^*\exp(\Delta) \\
\Rightarrow \exp(\Delta)(\eta^*-a(1-\eta^*)) &< 1-\eta^*(1+a).
\end{aligned}
\tag{34}
$$

We have $\eta^* - a(1-\eta^*) = \frac{\eta^*-\hat{\eta}}{1-\hat{\eta}} < 0$ and $1 - \eta^*(1+a) = 1 - \frac{\eta^*}{1-\hat{\eta}} > 0$. Therefore, Inequality 34 always holds true. $\square$

**Proof of Theorem 4.9**

*Proof.* Follow the definitions in (Bartlett et al., 2006), the generic conditional $\mathcal{L}$-risk is defined as $C_\eta(u) = (1 - \eta)\mathcal{L}(u) + \eta\mathcal{L}(-u)$. The optimal conditional $\mathcal{L}$-risk is defined as $H(\eta) = \inf_{u \in \mathbb{R}} C_\eta(u)$. Furthermore, we define $H^-(\eta) = \inf_{u(1-2\eta) \le 0} C_\eta(u)$ as the optimal risk under the constraint that the sign of $u$ is opposite to that of $1 - 2\eta$. In this scenario, we consider $0 \le \eta \le 1$ to ensure classification calibration while maintaining $0 \le a < 1$.

We have $C_\eta(u)$ for $\mathcal{L}_{\text{unbiased}}(u) = -\log\frac{\exp(u)+a}{\exp(u)+1}$ is:

$$
\begin{aligned}
C_\eta(u) &= (1 - \eta)[\log(1 + \exp(u)) - \log(a + \exp(u))] + \eta[\log(1 + \exp(-u)) - \log(a + \exp(-u))] \\
&= (1 - \eta)\log\frac{1 + \exp(u)}{a + \exp(u)} + \eta\log\frac{1 + \exp(u)}{1 + a\exp(u)} \\
&= \log(1 + \exp(u)) - (1 - \eta)\log(a + \exp(u)) - \eta\log(1 + a\exp(u))
\end{aligned}
\tag{35}
$$

Take the derivative of $u$ to 0:

$$
\frac{\partial C_\eta(u)}{\partial u} = \frac{\exp(u)}{1 + \exp(u)} - \frac{(1 - \eta)\exp(u)}{a + \exp(u)} - \frac{\eta a \exp(u)}{1 + a\exp(u)} = 0
\tag{36}
$$

Let $z = \exp(u)$, and then rearrange the equation:

$$
\begin{aligned}
\frac{1}{1 + z} &= \frac{1 - \eta}{a + z} + \frac{\eta a}{1 + az} \\
\Rightarrow (a + z)(1 + az) &= (1 - \eta)(1 + z)(1 + az) + \eta a(1 + z)(a + z) \\
\Rightarrow az^2 + (a^2 + 1)z + a &= az^2 + (a + 1)[1 + \eta(a - 1)]z + (1 - \eta + \eta a^2) \\
\Rightarrow z \cdot [-(1 - a)(a - \eta(a + 1))] &= (1 - a)[1 - \eta(1 + a)] \\
\Rightarrow z &= \frac{1 - \eta(1 + a)}{\eta(1 + a) - a}.
\end{aligned}
\tag{37}
$$

In order to ensure that $z > 0$ (which means that the $u$ has a real solution), we have $\frac{a}{1+a} < \eta < \frac{1}{1+a}$.

If $\frac{a}{1+a} < \eta < \frac{1}{1+a}$, substituting the optimal solution $z$, we have:

$$
\begin{aligned}
H(\eta) &= \eta[\log(1 + z) - \log(z + a)] + (1 - \eta)[\log(1 + z) - \log(1 + az)] \\
&= \mathcal{H}_{\text{bin}}(\eta) - \log(1 + a),
\end{aligned}
\tag{38}
$$

where $\mathcal{H}_{\text{bin}}(\eta) = -\eta\log\eta - (1 - \eta)\log(1 - \eta)$ is the binary entropy function.

If $\eta \le \frac{a}{1+a}$, the optimal solution is $u \to +\infty$, so we have $H(\eta) = -\eta\log a$.

If $\eta \ge \frac{1}{1+a}$, the optimal solution is $u \to -\infty$, so we have $H(\eta) = -(1 - \eta)\log a$.

For $\eta \le \frac{a}{1+a}$ and $\eta \ge \frac{1}{1+a}$, since $H(\eta)$ is symmetric with respect to $\eta = 1/2$, we can uniformly write it as:

$$
H(\eta) = -\min(\eta, 1 - \eta)\log a.
$$

For $H^-(\eta) = \inf_{u(1-2\eta) \le 0} C_\eta(u)$, we have $u \le 0$ when $\eta < \frac{1}{2}$, and $u \ge 0$ when $\eta > \frac{1}{2}$ Therefore, the constrained optimal solution must be achieved at the boundary $u = 0$. Thus, we have:

$$
H^-(\eta) = C_\eta(0) = \eta\phi(0) + (1 - \eta)\phi(0) = \phi(0) = \log\frac{2}{1 + a}.
\tag{39}
$$

Based on (Bartlett et al., 2006), A loss function is classification-calibrated if and only if for any $\eta \ne 1/2$, it holds that $H^-(\eta) > H(\eta)$.

In the middle region ($\frac{a}{1+a} < \eta < \frac{1}{1+a}$), the inequality $H^-(\eta) > H(\eta)$ becomes:

$$\log 2 - \log(1+a) > \mathcal{H}_{\text{bin}}(\eta) - \log(1+a) \iff \log 2 > \mathcal{H}_{\text{bin}}(\eta). \tag{40}$$

The binary entropy function $\mathcal{H}_{\text{bin}}(\eta)$ attains its maximum value of $\log 2$ at $\eta = 1/2$. Therefore, for $\eta \neq 1/2$, the inequality holds strictly.

In the boundary regions ($\eta \leq \frac{a}{1+a}$ and $\eta \geq \frac{1}{1+a}$), we observe that the boundary values are symmetric. Specifically, when $\eta = \frac{a}{1+a}$ or $\eta = \frac{1}{1+a}$, the function reaches a maximum of:$H(\eta) = -\frac{a}{1+a}\log a$. We must prove the following inequality:

$$\log \frac{2}{1+a} > -\frac{a}{1+a}\log a. \tag{41}$$

Define the auxiliary function $g(a) := \log \frac{2}{1+a} + \frac{a}{1+a}\log a$. Our objective is to show that $g(a) > 0$ for $a \in (0,1)$. We have $g'(a) = \frac{\log a}{(1+a)^2}$. Since $0 < a < 1$, it follows that $\log a < 0$, and consequently $g'(a) < 0$. This demonstrates that $g(a)$ is strictly monotonically decreasing on the interval $(0,1)$. Finally, we check the boundary condition at $a = 1$: $g(1) = \log \frac{2}{2} + \frac{1}{2}\log 1 = 0$. Because $a < 1$, we have $g(a) > 0$.

$\square$

**Proof of Corollary 4.10**

*Proof.* Define $\tilde{\psi}(\rho) = H^-(\frac{1+\rho}{2}) - H(\frac{1+\rho}{2})$, where $\rho \in [0,1]$. Since $H(\eta)$ is defined piecewise, $\tilde{\psi}(\rho)$ is also piecewise. Since $\frac{1+\rho}{2} \geq \frac{1}{2}$, we identify the transition point $\rho_0$ by setting $\frac{1+\rho_0}{2} = \frac{1}{1+a}$, which yields $\rho_0 = \frac{1-a}{1+a}$.

For $0 \leq \rho < \frac{1-a}{1+a}$, we have:

$$\tilde{\psi}(\rho) = (\log 2 - \log(1+a)) - (\mathcal{H}_{\text{bin}}(\eta) - \log(1+a)) = \log 2 - \mathcal{H}_{\text{bin}}\left(\frac{1+\rho}{2}\right). \tag{42}$$

For $\frac{1-a}{1+a} \leq \rho \leq 1$, we have:

$$\tilde{\psi}(\rho) = \log \frac{2}{1+a} - \left(-\frac{1-\rho}{2}\log a\right) = \log \frac{2}{1+a} + \frac{1-\rho}{2}\log a. \tag{43}$$

Convexity analysis: the first part $\log 2 - \mathcal{H}_{\text{bin}}(\frac{1+\rho}{2})$ is convex. The second part is linear (and also convex). At $\rho_0 = \frac{1-a}{1+a}$, the function values and first derivatives of the two parts are equal (continuous and smooth). Therefore, $\tilde{\psi}(\rho)$ is a convex function over the domain $[0,1]$. We define $\psi$ as the convex hull of $\tilde{\psi}$; thus, $\psi = \tilde{\psi}$.

According to Theorem 1 in (Bartlett et al., 2006), we have:

$$\psi(R(f) - R^*) \leq R_{\mathcal{L}_{\text{unbiased}}}(f) - R^*_{\mathcal{L}_{\text{unbiased}}}. \tag{44}$$

$\square$

# B. Experiments

## B.1. Experiment Details.

**Experiment Setting.** All experiments are based on Pytorch (Paszke et al., 2019) and TRL (von Werra et al., 2020) libraries, using 8 NVIDIA Pro 6000 (96GB) GPUs. For all the training, we use the AdamW optimizer, warmup ratio 0.1, batch size 128, maximum gradient norm 10. For SFT model training, we train the model for 1 epoch with learning rate 2e-5. For reward model training, we train the model for 3 epochs with learning rate 1e-5. For supervised preference training, we train the policy for 3 epochs with learning rate 5e-6 for Llama-3.2-3B and Qwen-3-1.7B, and 1e-6 for Llama-3.1-8B and Qwen-3-8B. For each dataset, we use the first 1000 different test samples as the test set.

**Baseline Hyperparameters.** For the regularization parameter $\beta$, following (Rafailov et al., 2023), we set it to 0.1 for the dialogue datasets HH and UFB, and 0.5 for the summarization dataset TL;DR. We conduct a hyperparameter search for each baseline on all noise level cases to ensure a completely fair experiment. For reward model training, we search $\epsilon \in \{0.1, 0.2, 0.4\}$ for cRM and rRM, and $a \in \{0.4, 0.6, 0.8\}$ for URM and $\alpha$-URM. For supervised preference training, we search $\epsilon \in \{0.1, 0.2, 0.4\}$ for cDPO and rDPO, $\beta' \in \{0.5, 1, 2\}$ for Dr.DPO, and $a \in \{0.4, 0.6, 0.8\}$ for UDPO and $\alpha$-UDPO. For practitioners, simply using $\alpha$-URM/$\alpha$-UDPO with $a = 0.8$ is usually sufficient to achieve strong performance.

### B.2. Additional Experiments

**Human Evaluation.** We conduct human evaluation on the case of Qwen-3-1.7B and HH with a 40% flip rate. We use the first 100 samples from the test set, and the human evaluation is independently performed by three annotators. The results are reported in Table 5. As shown, the human evaluation results are consistent with those obtained from GPT-5.

*Table 5.* Win rates (%) of different methods vs. the SFT model using GPT-5 and human judges. The top-2 best results are in bold.

| HH 40% | GPT-5 | Human |
|---|---|---|
| DPO | 62.0 | $62.6 \pm 2.8$ |
| **UDPO** | **71.0** | **$67.6 \pm 2.0$** |
| $\alpha$**-UDPO** | **68.0** | **$69.0 \pm 2.9$** |

**Instance-Dependent Noise.** We conduct experiments for instance-dependent noise. We first use a trained standard RM to obtain normalized reward scores for each sample in the HH dataset. For each sample, we define $\eta_x$ as $\frac{\min\{r(x,y_w), r(x,y_l)\}}{r(x,y_w) + r(x,y_l)}$. Intuitively, answer pairs with more similar reward scores are more likely to be mislabeled. We then flip labels according to $\eta_x$, with the noise rates rescaled to average corruption levels of 20% and 40%. The results using Qwen-3-1.7B are reported in Table 6. The results highlight the excellent performance of our methods under instance-dependent noise.

*Table 6.* Win rates (%) of different methods vs. the SFT model under instance-dependent noise. The top-2 best results are in bold.

| HH | IDN 20% | IDN 40% |
|---|---|---|
| RM | 56.8 | 46.0 |
| rRM | 58.0 | 48.2 |
| **URM** | **59.5** | **54.6** |
| $\alpha$**-URM** | **60.2** | **55.2** |

| HH | IDN 20% | IDN 40% |
|---|---|---|
| DPO | 68.7 | 56.4 |
| rDPO | 73.3 | 63.7 |
| **UDPO** | **77.9** | **68.0** |
| $\alpha$**-UDPO** | **78.2** | **67.6** |

**Closed Loop RLHF.** We evaluate the effectiveness of URM within the full RLHF loop. We use GRPO as the RL algorithm and conduct experiments using Qwen-3-1.7B on the HH dataset. We set the number of generations to 2 and train for 3 epochs. The results are reported in Table 7. These results demonstrate that our methods significantly improve the final RL policy performance in the closed loop RLHF.

*Table 7.* Win rates (%) of different methods using GRPO vs. the SFT model. The top-2 best results are in bold.

| GRPO | HH 0% | HH 20% | HH 40% |
|---|---|---|---|
| RM | 55.0 | 54.0 | 48.5 |
| **URM** | **60.3** | **58.7** | **56.5** |
| $\alpha$**-URM** | **61.6** | **58.2** | **57.0** |

**Larger Model.** We conduct an experiment using Qwen-3-14B on the UFB dataset, comparing our UDPO/$\alpha$-UDPO against standard DPO. The results are reported in Table 8. It can be observed that our methods remain highly effective on the larger model.

*Table 8.* Win rates (%) of different methods vs. the SFT model. The top-2 best results are in bold.

| UFB | Qwen-3-14B |
| --- | --- |
| DPO | 65.7 |
| **UDPO** | **73.0** |
| $\alpha$-**UDPO** | **73.8** |

