# OpenReview forum: "Unbiased Alignment for Large Language Models with Noisy Preferences"
_ICML.cc/2026/Conference — ICML 2026 regular_

### Official Review · Reviewer_TMMM · 2026-03-09

**Soundness:** 2
**Presentation:** 2
**Significance:** 2
**Originality:** 2
**Overall Recommendation:** 4
**Confidence:** 3

**Summary:**

This paper proposes an unbiased alignment framework for LLMs under noisy preference data. It models preference corruption as a label-flip transition process and analytically inverts it to derive two practical objectives: the Unbiased Reward Model (URM) loss for reward modeling in RLHF and the Unbiased Direct Preference Optimization (UDPO) loss for supervised preference learning (with normalized α variants). The method emphasizes noise tolerance and “parameter downward compatibility,” enabling robust training even when the true noise rate is unknown and the data may be cleaner than assumed. Experiments on HH, TL;DR, and UFB with additional manually flipped labels show consistent win-rate improvements over standard RM/DPO and strong robustness-oriented baselines, especially at higher noise levels. Overall, the work unifies noise-corrected reward and policy training while preserving the RLHF–DPO margin equivalence, supported by theoretical guarantees and extensive empirical validation.

**Compliance With Llm Reviewing Policy:**

Affirmed.

**Key Questions For Authors:**

1.Model scale. The current evaluation is conducted on relatively small models. Could the authors include 7B+ (and ideally larger) models to verify whether the same trends hold at scale?

2.Experiments. Beyond random label flips, could the authors evaluate on more realistic noise settings (e.g., instance-dependent/asymmetric/annotator-biased noise)? Also, since the reward model is ultimately used to train the RLHF policy, please provide at least brief end-to-end policy results to complete the RM→policy loop.

3.Noise modeling assumption: The paper adopts a simplified label-flip noise setting. Given that real-world preference noise is often more complex (e.g., instance-dependent, asymmetric, annotator-heterogeneous, or systematically biased), I encourage the authors to explain why they did not adopt or extend to more general noise models and to clarify the motivation and advantages of this assumption. It would also be helpful to explicitly state its applicability boundaries and provide preliminary empirical or theoretical support.

**Limitations:**

yes

**Strengths And Weaknesses:**

**Strengths**

It unifies reward learning in RLHF and policy learning in DPO under a single, implementable closed-form margin loss via noise correction, and provides an engineering-friendly downward-compatible tuning principle, enabling stable robustness gains even when the noise rate is unknown.


**Weaknesses**

1.The theoretical framework is built on a somewhat idealized noise assumption, whereas real-world noise is considerably more complex.

2.The “closed loop” on the RLHF side is incomplete: while URM is strong, it remains unclear whether the downstream gains on the final RL policy are sufficiently demonstrated.

---

> ### Author Rebuttal · Authors · 2026-03-30
>
> Thank you for your constructive comments and insightful suggestions.
>
> **Response to Noise Modeling Assumption (Weakness 1, Part of Question 2, and Question 3):**
> Because asymmetric and symmetric noise are equivalent in the alignment task, we mainly focus on instance-dependent noise. Our theory can naturally extend to instance-dependent noise. Specifically, let $\eta_x$ denote the noise rate for instance $x$, and define the corresponding constant $a_x$ as $\frac{\eta_x}{1-\eta_x}$.
> Since the proofs of the noise correction process in Theorem 4.2 and Theorem 4.5 are derived pointwise, the instance-dependent versions follow directly by replacing the constant $a$ with $a_x$.
> Although the $\eta_x$ of each instance is unknown, our parameter downward compatibility further implies that it suffices to use a single global parameter $\hat a$ satisfying $\hat a \ge \sup_x a_x$ (equivalently, $\hat{\eta} \ge \sup_x \eta_x$). In this case, the unbiased loss still recovers the unbiased optimal solution. We will revise the paper to incorporate the instance-dependent noise assumption and update the theoretical description accordingly.
>
> We also conduct experiments for instance-dependent noise. We first use a trained standard RM to obtain normalized reward scores for each sample in the HH dataset. For each sample, we define $\eta_x$ as
> $\frac{\min\\{r(x, y_w), r(x, y_l)\\}}{r(x, y_w) + r(x, y_l)}.$
> Intuitively, answer pairs with more similar reward scores are more likely to be mislabeled. We then flip labels according to $\eta_x$, with the noise rates rescaled to average corruption levels of 20\% and 40\%. The results using Qwen-3-1.7B are as follows:
>
> |HH|IDN 20\%|IDN 40\%|HH|IDN 20\%|IDN 40\%|
> |:-:|:-:|:-:|:-:|:-:|:-:|
> |RM|56.8|46.0|DPO|68.7|56.4|
> |rRM|58.0|48.2|rDPO|73.3|63.7|
> |**URM**|**59.5**|**54.6**|**UDPO**|**77.9**|**68.0**|
> |**$\alpha$-URM**|**60.2**|**55.2**|**$\alpha$-UDPO**|**78.2**|**67.6**|
>
> The results highlight the excellent performance of our methods under instance-dependent noise.
>
> **Response to The Closed Loop RLHF (Weakness 2 and Part of Question 2):**
> Following your suggestion, we evaluate the effectiveness of URM within the full RLHF loop. We use GRPO as the RL algorithm and conduct experiments using Qwen-3-1.7B on the HH dataset. The results are as follows:
>
> | GRPO  |HH 0\%|HH 20\%|HH 40\%|
> |:-----:|:----:|:----:|:----:|
> |   RM  | 55.0 | 54.0 | 48.5 |
> |**URM**|**60.3**|**58.7**|**56.5**|
> |**$\alpha$-URM**|**61.6**|**58.2**|**57.0**|
>
> These results demonstrate that our methods significantly improve the final RL policy performance in the closed loop RLHF.
>
> **Response to Model Scale (Question 1):**
> Following your suggestion, we conduct an experiment using Qwen-3-14B on the UFB dataset, comparing our UDPO/$\alpha$-UDPO against standard DPO. The results are as follows:
>
> |   UFB  | Qwen-3-14B |
> |:------:|:----------:|
> |   DPO|   65.7   |
> |**UDPO**|  **73.0** |
> |**$\alpha$-UDPO**|**73.8**|
>
> It can be observed that our methods remain highly effective on the larger model.

---

> > ### Author Rebuttal · Reviewer_TMMM · 2026-04-06
> >
> > I thank the authors for their rebuttal. It has addressed my main concerns,, which makes me stick to the original score.

---

> > > ### Author Response · Authors · 2026-04-06
> > >
> > > Thank you for approving our response. Your guidance has been instrumental in enhancing our work.

---

### Official Review · Reviewer_QyET · 2026-03-11

**Soundness:** 2
**Presentation:** 3
**Significance:** 2
**Originality:** 2
**Overall Recommendation:** 3
**Confidence:** 4

**Summary:**

This paper studies alignment under noisy preference data. It proposes two objectives, URM for reward modeling and UDPO for direct preference optimization, by modeling noisy preferences as a corruption process and deriving corrected losses. The paper also provides theoretical analysis on gradient behavior, downward compatibility, and classification calibration, and reports experiments on HH, TL;DR, and UFB showing gains over several noisy preference baselines.

**Compliance With Llm Reviewing Policy:**

Affirmed.

**Final Justification:**

This paper addresses an important problem, namely alignment under noisy preference data, and proposes a simple framework that covers both reward modeling and direct preference optimization. I appreciate the combination of theory and experiments, and I find the robustness motivation meaningful.

My main concern, however, is only partially resolved by the rebuttal. The method is presented as “unbiased,” but in practice it still depends on choosing a noise-related hyperparameter when the true noise rate is unknown. The rebuttal improved my view by adding instance-dependent noise results, additional UFB stability evidence across multiple seeds and judges, and clarifying which claims are theoretical. This was enough for me to increase my score by one notch. Still, the theory-to-practice gap remains, and I also remain concerned about presentation and reproducibility issues such as the inconsistent base model name.

**Key Questions For Authors:**

1. The theory defines $ a $ from the true noise rate, but the experiments tune $a $. How should a practitioner choose $ a $ in a realistic setting where the noise rate is unknown and no clean validation signal is available? A convincing answer would improve my view of the practical value of the method.

2.  Can the authors provide stronger evidence on real or structured noise, beyond random label flips? For example, instance dependent or annotator specific noise would make the robustness claim more convincing.

3.  On UFB, the gains over strong baselines are relatively small. Are these improvements stable across random seeds and judge models? If yes, that would strengthen the empirical case.

4. The paper claims an unbiased solution under noisy preferences. Could the authors clarify more explicitly which claims are theoretical?

5. Why do the ablation experiments involve Llama-3.1-3B, whereas Section 5.1 utilizes Llama-3.2-3B? Does Llama-3.1-3B exist?

**Limitations:**

No. The discussion of limitations is too brief. The paper should more clearly acknowledge that the practical method still depends on choosing a noise related parameter, that much of the robustness evidence comes from synthetic label flips, and that robustness to more realistic noise patterns is not evaluated.

**Strengths And Weaknesses:**

Strengths:
1. The paper addresses an important problem. Noisy preference data is common in practical alignment, and improving robustness in this setting is meaningful.

2. The method is simple and easy to follow. Extending the framework to both reward modeling and direct preference optimization is a nice design choice.

3. The paper includes both theory and experiments. The empirical study covers multiple datasets, several baselines, and both reward model training and policy optimization.

4. The results are generally strong. UDPO and its normalized version outperform DPO based baselines in most settings, especially when additional label flips are introduced.

Weaknesses:

1. My main concern is the gap between the theory and the practical method. The core correction uses $ a = \eta / (1 - \eta) $, but the paper also states that the true noise rate is unobservable in practice and therefore treats (a) as a tunable hyperparameter. This weakens the central claim of unbiased alignment, since the practical recipe still depends on selecting a noise related parameter.

2. The paper argues that downward compatibility makes the method practical, but the experiments still rely on searching over several values of $ a $. It is therefore not fully clear how the method should be used when the noise level is unknown and no tuning signal is available.

3. A large part of the empirical evidence comes from manually flipped labels. This is useful, but synthetic random flips are much simpler than real preference noise such as annotator bias, ambiguity, or instance dependent corruption. The paper does not show whether the method remains effective under more realistic noise patterns.

4. The gains on the harder UFB setting are modest. For supervised preference learning, the improvement over the strongest baselines is around 2 points, which makes the practical advantage less convincing than the headline claims.

5. The ablation section refers to Llama-3.1-3B while Section 5.1 uses Llama-3.2-3B.

6. There are several typographical errors, such as "as as following".

---

> ### Author Rebuttal · Authors · 2026-03-30
>
> Thank you for your constructive comments and insightful suggestions.
>
> **Response to Weakness 1/2 and Question 1:**
> Our methods have parameter downward compatibility. However, $a$ scales the gradients of URM/UDPO, so different $a$ may require different learning rates. Since we use a fixed learning rate in experiments, varying $a$ can affect the performance of URM/UDPO. To address this issue, we further propose normalized $\alpha$-URM/$\alpha$-UDPO, which require only a fixed choice of $a = 0.8$ to obtain strong  results, without the need for parameter selection. This is consistent with our theory of parameter downward compatibility, and the ablation results in Figure 3 verify this property. **For practitioners, simply using $\alpha$-URM/$\alpha$-UDPO with $a = 0.8$ is sufficient to achieve strong performance.** We will add this clearer explanation in the final version.
>
> **Response to Weakness 3 and Question 2:**
> Our theory can naturally extend to instance-dependent noise. Specifically, let $\eta_x$ denote the noise rate for instance $x$, and define the corresponding constant $a_x$ as $\frac{\eta_x}{1-\eta_x}$. Since the proofs of the noise correction process in Theorem 4.2 and Theorem 4.5 are derived pointwise, the instance-dependent versions follow directly by replacing the constant $a$ with $a_x$. Although the $\eta_x$ of each instance is unknown, our parameter downward compatibility further implies that it suffices to use a single global parameter $\hat a$ satisfying $\hat a \ge \sup_x a_x$ (equivalently, $\hat{\eta} \ge \sup_x \eta_x$). In this case, the unbiased loss still recovers the unbiased optimal solution. We will revise the paper to incorporate the instance-dependent noise assumption and update the theoretical description accordingly.
>
> We also conduct experiments for instance-dependent noise. We first use a trained standard RM to obtain normalized reward scores for each sample in the HH dataset. For each sample, we define $\eta_x$ as $\frac{\min\\{r(x, y_w), r(x, y_l)\\}}{r(x, y_w) + r(x, y_l)}.$ Intuitively, answer pairs with more similar reward scores are more likely to be mislabeled. We then flip labels according to $\eta_x$, with the noise rates rescaled to average corruption levels of 20\% and 40\%. The results using Qwen-3-1.7B are as follows:
>
> |HH|IDN 20\%|IDN 40\%|HH|IDN 20\%|IDN 40\%|
> |:-:|:-:|:-:|:-:|:-:|:-:|
> |RM|56.8|46.0|DPO|68.7|56.4|
> |rRM|58.0|48.2|rDPO|73.3|63.7|
> |**URM**|**59.5**|**54.6**|**UDPO**|**77.9**|**68.0**|
> |**$\alpha$-URM**|**60.2**|**55.2**|**$\alpha$-UDPO**|**78.2**|**67.6**|
>
> The results highlight the excellent performance of our methods under instance-dependent noise.
>
> **Response to Weakness 4 and Question 3:**
> For UFB, we use the original dataset without additional manual noise, so the performance gaps among baselines are relatively small. Following your suggestion, we conduct 3 random runs for all UFB experiments and use two judges, GPT-5 and Claude-Opus-4.6, to evaluate the results, respectively. The results are as follows:
>
> |GPT-5|Llama-3.1-8B|Qwen-3-8B|Claude-4.6|Llama-3.1-8B|Qwen-3-8B|
> |:-:|:-:|:-:|:-:|:-:|:-:|
> |RM|67.6±1.2|55.1±0.7|RM|66.7±1.3|52.6±0.4|
> |cRM|66.1±1.6|52.3±2.0|cRM|62.4±0.7|50.6±1.9|
> |rRM|64.2±0.5|63.3±0.9|rRM|66.2±1.2|62.5±1.0|
> |**URM**|**71.8±1.0**|**65.1±1.6**|**URM**|**70.6±1.2**|**63.8±0.8**|
> |**$\alpha$-URM**|**72.4±0.4**|**64.7±0.9**|**$\alpha$-URM**|**71.7±1.0**|**64.2±1.3**|
>
> |GPT-5|Llama-3.1-8B|Qwen-3-8B|Claude-4.6|Llama-3.1-8B|Qwen-3-8B|
> |:-:|:-:|:-:|:-:|:-:|:-:|
> |DPO|66.8±1.0| 56.4±2.0|DPO|63.6±1.1|53.7±1.1|
> |cDPO|64.7±1.2| 55.0±1.6|cDPO|64.1±0.8|54.0±1.4|
> |IPO|72.2±0.8| 57.6±1.2|IPO|68.1±0.7|53.9±0.9|
> |rDPO|72.3±1.2| 58.1±1.3|rDPO|67.2±0.9|55.9±0.9|
> |Dr.DPO|66.2±1.2| 56.5±0.4|Dr.DPO|64.7±2.0|54.8±0.8|
> |**UDPO**|**74.1±1.6**|**61.2±1.3**|**UDPO**|**71.2±0.5**|**58.2±1.6**|
> |**$\alpha$-UDPO**|**73.3±1.0**|**60.1±0.7**|**$\alpha$-UDPO**|**70.5±1.2**|**58.6±0.4**|
>
> The results produced by different judges are consistent, and our methods stably achieve the best performance.
>
> **Response to Question 4:**
> We clarify that our claim about the unbiased solution is theoretical. Specifically, Theorems 4.2 and 4.5 establish the noise correction process; Corollaries 4.3 and 4.6 show that URM/UDPO have the clean optimum under noisy training; and Corollary 4.7 gives the downward compatibility property. In contrast, the performance gains in Tables 1–4 with performance improvements are empirical results. We will revise the paper to make the claim more explicit.
>
> **Response to Weakness 5/6 and Question 5:**
> The "Llama-3.1-3B" in the ablation section is a typo.  We use Llama-3.2-3B in this paper. We sincerely apologize for this typo. We have thoroughly reviewed the manuscript and corrected all typographical errors, such as “as as following.” Thank you for your reminder.
>
> **Response to Limitation:**
> Regarding the concerns raised in Limitation, we have answered them in our above response. We hope our explanation will resolve your concerns.

---

> > ### Author Rebuttal · Reviewer_QyET · 2026-04-03
> >
> > Thank you for the detailed rebuttal and the additional experiments. The new instance-dependent noise experiment, the extra UFB results across multiple seeds and two judges, and the clarification of which claims are theoretical all improve my assessment. I am therefore increasing my score from 2 to 3.
> >
> > I remain concerned about the paper’s presentation and reproducibility. In particular, the inconsistency in the reported base model name (Llama-3.1-3B vs. Llama-3.2-3B) is not a minor typo, since it directly affects the clarity and reproducibility of the experimental setup. Combined with the remaining practical gap, this prevents me from raising my score further.
> >
> > Overall, the rebuttal improves my view of the paper and justifies a one-notch increase, but my core concerns are not fully resolved.

---

> > > ### Author Response · Authors · 2026-04-03
> > >
> > > Thank you for taking the time to reply. We would like to further clarify the remaining concerns.
> > >
> > > 1) On the textual error: We agree that precise model names are important for clarity and reproducibility, and we sincerely apologize for the typo "Llama-3.1-3B" in Line 414. In reality, Llama-3.1-3B does not exist; the model we use is Llama-3.2-3B. The model names in other places are correct, and we have corrected this typo. This is a textual error; in actual experiments, there are no inconsistencies or reproducibility issues. After your reminder and our correction, we believe this will not affect the readers' reading. Thank you again for your reminder.
> > >
> > > 2) On the practical gap: The sensitivity of URM/UDPO to $a$ mainly comes from the fact that $a$ changes the gradient scale, so different $a$ may require different learning rates for best performance. In our experiments, we did not change the learning rate. To address this issue, we propose normalized $\alpha$-URM/$\alpha$-UDPO: with a fixed choice of $a = 0.8$, **they can achieve strong results without parameter tuning**. This is consistent with our theory. The search over $a$ is included for sensitivity analysis. We agree that the practical guidance should be made clearer, and we have revised the manuscript accordingly to improve both the explanation and the presentation.
> > >
> > > We hope this clarification addresses your remaining concerns. We invest substantial time and effort in preparing this paper, and we sincerely regret that this inadvertent mistake may have caused unnecessary concern. We would be grateful if you would reconsider our work.

---

### Official Review · Reviewer_nNLw · 2026-03-13

**Soundness:** 4
**Presentation:** 4
**Significance:** 4
**Originality:** 4
**Overall Recommendation:** 5
**Confidence:** 4

**Summary:**

This paper addresses the problem of training LLMs from noisy preference data, where a fraction η of preference labels are flipped. The authors model noise as symmetric label flipping (Eq. 6), derive the mathematical relationship between noisy and clean reward/policy models, and invert it to obtain corrected loss functions: the Unbiased Reward Model (URM) loss for RLHF and the Unbiased Direct Preference Optimization (UDPO) loss for DPO. Both reduce to a single unified form L = -log((exp(Δ)+a)/(exp(Δ)+1)), where a = η/(1-η) and Δ is the reward margin. The paper provides theoretical analysis including gradient behavior, parameter downward compatibility (overestimating noise is safe), and classification calibration (minimizing the loss recovers the Bayes-optimal classifier). Experiments on three datasets (HH, TL;DR, UFB) with four models show consistent improvements over baselines, with particularly large gains under high noise.

**Compliance With Llm Reviewing Policy:**

Affirmed.

**Key Questions For Authors:**

* How robust is the method to asymmetric or instance-dependent noise? Even a synthetic experiment where hard pairs (small true reward gap) are flipped at higher rates than easy pairs would be informative.

* Can you disentangle the noise-correction effect from generic regularization? The 0% flip rate gains suggest a helps even on relatively clean data. An experiment comparing UDPO against a simple regularized DPO variant with a matched hyperparameter budget would clarify this.
* Can you provide human evaluation, even at small scale? Given the paper is about noise in human preferences, verifying that the GPT-5 judge isn't systematically biased toward one method is important.
* How does UDPO compare to data filtering approaches (e.g., removing pairs where annotators disagree or where reward model confidence is low)?

**Limitations:**

The symmetric noise assumption is acknowledged implicitly but not discussed as a limitation. The paper would benefit from explicitly stating where the framework might break down (asymmetric noise, adversarial annotation, instance-dependent noise rates) and discussing the relationship between noise correction and regularization.

**Strengths And Weaknesses:**

Strengths

* Elegant unified framework. The noise correction yields one loss function with one parameter a that works for both reward models and DPO, and preserves the RLHF-DPO equivalence. The derivation (Theorems 4.2, 4.5) is clean — invert the noise-to-clean sigmoid distortion algebraically.
* Gradient analysis is the key insight (Section 4.3). Standard DPO/RM loss gives the largest gradient to samples with highly negative Δ, which are disproportionately mislabeled. UDPO's gradient vanishes for those samples and peaks at Δ = ½ log(a), focusing optimization on genuinely ambiguous pairs. This explains mechanistically why DPO overfits noise and why the correction works.
* Parameter downward compatibility (Corollary 4.7). If you overestimate the noise rate, the loss remains unbiased. This is practically important since η is unknown, and it distinguishes UDPO from rDPO and label smoothing, where overestimation degrades performance. The proof is clean and the ablation (Figure 3) confirms it empirically.
* Classification calibration and excess risk bound (Theorem 4.9, Corollary 4.10). The loss recovers the Bayes-optimal classifier, with a piecewise risk bound that behaves quadratically near the optimum and linearly in the high-error regime. This provides solid theoretical grounding.
* Comprehensive experiments. Four models (3B and 8B scale), three datasets, both RM and DPO settings, controlled noise at 0%/20%/40%, and relevant baselines (rDPO, Dr.DPO, IPO, label smoothing). Gains are substantial (~15% over DPO, ~4% over SOTA on average). The synthetic validation (Figure 2) cleanly confirms the theory.

Weaknesses
* Symmetric noise model is a strong assumption. Equation 6 assumes a uniform flip rate η across all pairs. Real annotation noise is instance-dependent — ambiguous pairs (close in quality) are noisier than clear-cut ones, and different annotators have different error profiles. The paper never discusses robustness to asymmetric or instance-dependent noise. To be clear, the paper framing is justified and assuming symmetric noise is reasonable for deriving an elegant framework + algorithm, but some experiments showing whether the method is effective in instances of asymmetric noise would be very interesting to see.
* In practice, a is a tuned hyperparameter. Since η is unknown, a is searched over a grid ({0.4, 0.6, 0.8} for UDPO). The downward compatibility property justifies why this is safe, but it blurs the distinction between "principled noise correction" and "a helpful regularization knob." The paper should be more transparent about this.
* The 0% flip rate gains need more discussion. URM/UDPO outperform baselines even without manual noise injection (Tables 1-2), attributed to "intrinsic noise." But this suggests a is also acting as a general regularizer beyond noise correction. If the loss helps even on relatively clean data, what exactly is the noise correction buying over generic regularization? An experiment disentangling these effects would strengthen the contribution.
* Missing comparison with data filtering. If noisy pairs are the problem, an obvious baseline is identifying and removing them (e.g., via annotator disagreement or reward model confidence). This comparison would clarify whether the loss-correction approach is superior to data-centric approach

---

> ### Author Rebuttal · Authors · 2026-03-30
>
> Thank you for your constructive comments and insightful suggestions.
>
> **Response to Weakness 1 and Question 1:**
> Because asymmetric and symmetric noise are equivalent in the alignment task, we mainly focus on instance-dependent noise. Our theory can naturally extend to instance-dependent noise. Specifically, let $\eta_x$ denote the noise rate for instance $x$, and define the corresponding constant $a_x$ as $\frac{\eta_x}{1-\eta_x}$.
> Since the proofs of the noise correction process in Theorem 4.2 and Theorem 4.5 are derived pointwise, the instance-dependent versions follow directly by replacing the constant $a$ with $a_x$.
> Although the $\eta_x$ of each instance is unknown, our parameter downward compatibility further implies that it suffices to use a single global parameter $\hat a$ satisfying $\hat a \ge \sup_x a_x$ (equivalently, $\hat{\eta} \ge \sup_x \eta_x$). In this case, the unbiased loss still recovers the unbiased optimal solution. We will revise the paper to incorporate the instance-dependent noise assumption and update the theoretical description accordingly.
>
> We also conduct experiments for instance-dependent noise. We first use a trained standard RM to obtain normalized reward scores for each sample in the HH dataset. For each sample, we define $\eta_x$ as
> $\frac{\min\\{r(x, y_w), r(x, y_l)\\}}{r(x, y_w) + r(x, y_l)}.$
> Intuitively, answer pairs with more similar reward scores are more likely to be mislabeled. We then flip labels according to $\eta_x$, with the noise rates rescaled to average corruption levels of 20\% and 40\%. The results using Qwen-3-1.7B are as follows:
>
> |HH|IDN 20\%|IDN 40\%|HH|IDN 20\%|IDN 40\%|
> |:-:|:-:|:-:|:-:|:-:|:-:|
> |RM|56.8|46.0|DPO|68.7|56.4|
> |rRM|58.0|48.2|rDPO|73.3|63.7|
> |**URM**|**59.5**|**54.6**|**UDPO**|**77.9**|**68.0**|
> |**$\alpha$-URM**|**60.2**|**55.2**|**$\alpha$-UDPO**|**78.2**|**67.6**|
>
> The results highlight the excellent performance of our methods under instance-dependent noise.
>
> **Response to Weakness 2:**
> 1) For hyperparameter $a$: We present an ablation experiment for $a$ in Figure 3.
> URM/UDPO are somewhat sensitive to $a$ because $a$ alters the scale of the gradient. We propose normalized $\alpha$-URM/$\alpha$-UDPO to address this issue: with a fixed choice of $a = 0.8$, it can achieve strong results without parameter tuning.
> 2) For noise correction and regularization: we believe that the gains of our methods over standard RM/DPO mainly stem from the principled noise correction based on parameter $a$.
> As for regularization, our method inherits the same regularization mechanism as standard DPO, and we fix the parameter $\beta$ to the same value used in DPO.
>
> **Response to Weakness 3 and Question 2:**
> We believe that the improvements of our methods on the original datasets still stem from their noise-tolerant design. This is because real-world datasets inherently contain substantial noise.
> Generic regularization cannot effectively mitigate noisy preferences, as standard DPO inherently possesses regularization capabilities through the parameter $\beta$.
> Moreover, the baselines cDPO and IPO are both variants designed to further strengthen DPO’s generic regularization capability, yet our methods still significantly outperform them.
>
> **Response to Question 3:**
> We don't have enough time to complete the human evaluation within the rebuttal window.
> The previous study [1] has shown that human judge and advanced LLM judge are highly consistent. To strengthen the evaluation evidence, we conduct 3 random runs for all UFB experiments and use two judges, GPT-5 and Claude-Opus-4.6, to evaluate the results, respectively.
> Due to the response-length limit, please see experiment results in "Response to Weakness 4 and Question 3." of the rebuttal to reviewer QyET.
>
> It can be seen that the results produced by different judges are consistent, and our methods stably achieve the best performance.
>
> [1] Direct Preference Optimization: Your Language Model is Secretly a Reward Model, NeurIPS 2023.
>
> **Response to Weakness 4 and Question 4:**
> We conduct an experiment for a simple data filtering method. Specifically, we first train the model for one epoch using standard DPO, then remove samples with high loss values, and finally continue the training process on the remaining data. The filtering rate is set to 10\%, 20\%, and 40\% for 0\%, 20\%, and 40\% label flipping settings, respectively. We perform experiments with Qwen-3-1.7B on the HH dataset, and the results are as follows:
>
> |HH|0\%|20\%|40\%|
> |:-:|:-:|:-:|:-:|
> |DPO|75.4|68.2|58.2|
> |DPO (filtering)|77.8|70.4|63.7|
> |**UDPO**|**83.4**|**81.2**|**73.0**|
> |**$\alpha$-UDPO**|**81.3**|**80.3**|**73.2**|
>
> It can be seen that the data filtering method improves over standard DPO, but its performance is still not competitive.
>
> **Response to Limitation:**
> Regarding the concerns raised in Limitation, we have answered them in our above response. We hope our explanation will resolve your concerns.

---

> > ### Author Rebuttal · Reviewer_nNLw · 2026-03-31
> >
> > Thank you for taking the time to write your rebuttal with additional results. I enjoyed reading this paper and will maintain my score of 5 (accept).
> >
> > For the full paper, a human evaluation would be good. Even a small sample of humans would provide useful signal.

---

> > > ### Author Response · Authors · 2026-04-01
> > >
> > > Thank you for your reply and approval. We will include human evaluation in the final version. Your review has played a crucial role in improving our work.

---

### Official Review · Reviewer_AcEs · 2026-03-24

**Soundness:** 3
**Presentation:** 3
**Significance:** 3
**Originality:** 2
**Overall Recommendation:** 4
**Confidence:** 3

**Summary:**

This paper studies LLM alignment under noisy pairwise preferences. The core idea is to model preference corruption as symmetric label-flip noise and derive corrected objectives that recover the underlying clean preference signal from noisy labels. The paper proposes two methods: Unbiased RM (URM) for reward modeling in RLHF, and Unbiased DPO (UDPO) for direct preference learning. Intuitively, the corrected objective down-weights pairs where the model strongly disagrees with the observed preference, making training less sensitive to potentially noisy labels. The paper further shows that the two objectives admit a unified form and preserve the usual connection between RLHF and DPO. On the theory side, it provides analysis of noise tolerance, downward compatibility, and classification calibration. Empirically, the method is evaluated on several standard preference datasets and shows improvements over standard RM/DPO and several robust preference-learning baselines under both native and synthetically corrupted preference noise.

**Compliance With Llm Reviewing Policy:**

Affirmed.

**Key Questions For Authors:**

Please refer to the weaknesses above. In addition, I have the following question:

1. (Regarding W2) Could the authors add an instance-dependent noise experiment, e.g., by using an off-the-shelf reward model or LLM judge to identify ambiguous low-margin pairs and assigning them higher flip rates than easy pairs? Since the current synthetic corruption matches the symmetric-flip assumption closely, this would better test robustness under more realistic preference noise.

**Limitations:**

The paper could strengthen its limitation discussion in the following respects:

1. The analysis is restricted to symmetric label noise, which may not reflect realistic preference noise in practice, where ambiguity and instance-dependent disagreement are common.
2. The paper does not discuss sensitivity to the hyperparameter $a$ even though $a$ plays a central role in controlling the degree of noise correction.

**Strengths And Weaknesses:**

This work has some notable strengths. Most importantly, the unified formulation of URM and UDPO is elegant: the two objectives are derived from different starting points but reduce to the same margin-based loss, yielding a clean connection between RLHF and DPO under noisy preferences. The downward compatibility result is also practically valuable, since it shows that overestimating the noise level does not harm optimality when the data is cleaner than assumed. In addition, the gradient analysis provides clear intuition for robustness by showing that the loss down-weights extreme disagreements, which are more likely to correspond to noisy labels under the assumed model. Finally, the empirical evaluation is reasonably comprehensive and shows consistent improvements over standard and robust baselines across several datasets and model families.

Beyond these strengths, I have the following concerns and questions:

1. The paper appears to overstate the novelty of its core loss derivation. Based on my understanding, the proposed URM/UDPO objective is closely related to classical loss-correction methods under symmetric label noise. The paper assumes the standard symmetric flip model and derives a corrected Bradley–Terry / DPO-style loss, while rDPO can be viewed as a different correction of the same underlying noisy-preference objective. In that sense, the main novelty seems to lie less in the corrected loss itself, and more in its extension to reward-model training, the downward-compatibility result, and the practical treatment of the noise parameter. Given that the introduction frames this as a “brand-new unbiased alignment theoretical framework,” the paper should position itself much more carefully relative to the noisy-label literature and robust preference-learning work.

2. The method relies on a symmetric global noise-rate assumption, which is likely too restrictive for real preference data. In practice, noise is often instance-dependent, with ambiguous pairs being much noisier than easy ones. Since the synthetic label-flip experiments closely match the assumed noise model, the paper does not test robustness under more realistic misspecification.

3. The treatment of the noise hyperparameter $a$ is under-documented. The paper does not clearly report how $a$ is selected in each setting, or whether it is tuned separately for each noise level. Since the base datasets already contain natural noise, URM/UDPO with $a>0$ has an extra noise-correction degree of freedom even at 0% synthetic corruption, which makes the comparison to standard RM/DPO somewhat difficult to interpret.

---

> ### Author Rebuttal · Authors · 2026-03-30
>
> Thank you for your constructive comments and insightful suggestions.
>
> **Response to Weakness 1:**
> While prior studies on robust alignment and symmetric loss share our goal of mitigating noise, our approach differs fundamentally in its derivation. We develop our method independently, without relying on these existing works. Consequently, our method offers several advantageous properties compared with previous approaches, such as parameter backward compatibility. We sincerely appreciate your suggestion. We will introduce our work more appropriately and include a broader discussion of the literature on noisy labels and robust preference learning in both the Introduction and Related Work sections.
>
> **Response to Weakness 2, Question 1, and Limitation 1:**
> Our theory can naturally extend to instance-dependent noise. Specifically, let $\eta_x$ denote the noise rate for instance $x$, and define the corresponding constant $a_x$ as $\frac{\eta_x}{1-\eta_x}$.
> Since the proofs of the noise correction process in Theorem 4.2 and Theorem 4.5 are derived pointwise, the instance-dependent versions follow directly by replacing the constant $a$ with $a_x$.
> Although the $\eta_x$ of each instance is unknown, our parameter downward compatibility further implies that it suffices to use a single global parameter $\hat a$ satisfying $\hat a \ge \sup_x a_x$ (equivalently, $\hat{\eta} \ge \sup_x \eta_x$). In this case, the unbiased loss still recovers the unbiased optimal solution. We will revise the paper to incorporate the instance-dependent noise assumption and update the theoretical description accordingly.
>
> We also conduct experiments for instance-dependent noise. We first use a trained standard RM to obtain normalized reward scores for each sample in the HH dataset. For each sample, we define $\eta_x$ as
> $\frac{\min\\{r(x, y_w), r(x, y_l)\\}}{r(x, y_w) + r(x, y_l)}.$
> Intuitively, answer pairs with more similar reward scores are more likely to be mislabeled. We then flip labels according to $\eta_x$, with the noise rates rescaled to average corruption levels of 20\% and 40\%. The results using Qwen-3-1.7B are as follows:
>
> |HH|IDN 20\%|IDN 40\%|HH|IDN 20\%|IDN 40\%|
> |:-:|:-:|:-:|:-:|:-:|:-:|
> |RM|56.8|46.0|DPO|68.7|56.4|
> |rRM|58.0|48.2|rDPO|73.3|63.7|
> |**URM**|**59.5**|**54.6**|**UDPO**|**77.9**|**68.0**|
> |**$\alpha$-URM**|**60.2**|**55.2**|**$\alpha$-UDPO**|**78.2**|**67.6**|
>
> The results highlight the excellent performance of our methods under instance-dependent noise.
>
>
> **Response to Weakness 3 and Limitation 2:**
> 1) The treatment and sensitivity of hyperparameter $a$:
> We present an ablation experiment for $a$ in Figure 3 of the paper.
> Specifically, $a$ scales the gradients of the URM/UDPO, so different $a$ may require different learning rates. Since we use a fixed learning rate in experiments, varying $a$ can affect the performance of URM/UDPO.
> Therefore, similar to the baselines, we tune $a$ separately for each noise level for URM/UDPO.
> To address this issue, we further propose normalized $\alpha$-URM/$\alpha$-UDPO, which require only a fixed choice of $a = 0.8$ and achieve strong results without hyperparameter tuning, consistent with our parameter downward compatibility theory.
> We will provide more detailed information on the selection of $a$ in the final version.
>
> 2) Comparison with standard RM/DPO on original datasets: Since real-world datasets are inherently noisy, mitigating preference noise is critical. This explains why our URM/UDPO significantly outperform standard RM/DPO even on the original dataset. The core advantage of our approach is its robustness against preference noise. In the idealized case of perfectly clean preferences (which is hard to achieve in practice), we believe URM/UDPO will perform comparably to standard RM/DPO. If we set $a = 0$, URM/UDPO reduce to standard RM/DPO.

---

> > ### Author Rebuttal · Reviewer_AcEs · 2026-04-04
> >
> > Thank you for the detailed rebuttal. I appreciate the additional experiment on instance-dependent noise. I keep my positive score.

---

> > > ### Author Response · Authors · 2026-04-04
> > >
> > > Thank you for approving our response. Your guidance has been instrumental in enhancing our work.

---

### Decision · Program_Chairs · 2026-04-30

**Decision:**

Accept (regular)

**Comment:**

This work introduces corrections to the reward loss function in RLHF as well as the loss function in DPO in order to achieve better robustness against label noise in the comparison data. Reviewers generally agree that this work makes solid contributions through theoretical analyses and empirical evaluations. There are concerns about the simplistic noise assumption (symmetric flips) and the challenge in choosing the hyper-parameter, i.e., the noise level parameter. The authors responded with additional results on instance-dependent noise. I trust the authors to implement the revision promised in the rebuttal. I recommend acceptance.